DOI: 10.1038/s41467-018-06744-7　　**OPEN**

# A phosphatidylinositol 4,5-bisphosphate redistribution-based sensing mechanism initiates a phagocytosis programing

Libing Mu[1,2], Zhongyuan Tu[3], Lin Miao[2,4], Hefei Ruan[1,2], Ning Kang[1,2], Yongzhen Hei[2], Jiahuan Chen[1,2], Wei Wei[5], Fangling Gong[5], Bingjie Wang[6], Yanan Du[6], Guanghui Ma[5], Matthias W. Amerein[7,8], Tie Xia[1,2] & Yan Shi[1,2,3]

Phagocytosis is one of the earliest cellular functions, developing approximately 2 billion years ago. Although FcR-based phagocytic signaling is well-studied, how it originated from ancient phagocytosis is unknown. Lipid redistribution upregulates a phagocytic program recapitulating FcR-based phagocytosis with complete dependence on Src family kinases, Syk, and phosphoinositide 3-kinases (PI3K). Here we show that in phagocytes, an atypical ITAM sequence in the ancient membrane anchor protein Moesin transduces signal without receptor activation. Plasma membrane deformation created by solid structure binding generates phosphatidylinositol 4,5-bisphosphate (PIP2) accumulation at the contact site, which binds the Moesin FERM domain and relocalizes Syk to the membrane via the ITAM motif. Phylogenic analysis traces this signaling using PI3K and Syk to 0.8 billion years ago, earlier than immune receptor signaling. The proposed general model of solid structure phagocytosis implies a preexisting lipid redistribution-based activation platform collecting intracellular signaling components for the emergence of immune receptors.

[1] School of Life Sciences, Tsinghua University, Beijing 100084, China. [2] Institute for Immunology and Department of Basic Medical Sciences, Beijing Key Lab for Immunological Research on Chronic Diseases, School of Medicine; Tsinghua-Peking Center for Life Sciences, Tsinghua University, Beijing 100084, China. [3] Department of Microbiology, Immunology & Infectious Diseases and Snyder Institute, University of Calgary, Calgary T2N 4N1 AB, Canada. [4] School of Life Sciences, Peking-Tsinghua Center for Life Sciences, Peking University, Beijing 100871, China. [5] State Key Laboratory of Biochemical Engineering, Institute of Process Engineering, Chinese Academy of Sciences, Beijing 100190, China. [6] Department of Biomedical Engineering, School of Medicine, Collaborative Innovation Center for Diagnosis and Treatment of Infectious Diseases, Tsinghua University, Beijing 100084, China. [7] Department of Cell Biology and Anatomy, University of Calgary, Calgary T2N 4N1 AB, Canada. [8] Snyder Institute of Chronic Diseases, University of Calgary, Calgary T2N 4N1 AB, Canada. These authors contributed equally: Libing Mu, Zhongyuan Tu. Correspondence and requests for materials should be addressed to T.X. (email: xiatie@biomed.tsinghua.edu.cn) or to Y.S. (email: yanshi@biomed.tsinghua.edu.cn)

Phagocytosis is the eat to defend functionalization of the original eat to feed feature in the primordial eukaryotic life forms[1], and the conceptual foundation of cellular immunity[2]. It started at the eukaryogenesis about two billion years ago[3]. In single cell life forms, phagocytosis endows energy/nutrient harvesting. In modern immune cells, phagocytosis has become a functional specialty in a subset of cells[4]. Although phagocytic signaling in immune cells has been mostly delineated[5], it is unknown how it originated from the ancient architype.

Immune receptors share a common signaling cascade[6]. A receptor ligation leads to the phosphorylation of intracellular ITAM motif. ITAM dual YXXL sequences then bind to SH2 present in spleen tyrosine kinase (Syk) or ζ-chain-associated protein kinase 70 (ZAP70)[7]. Syk then recruits downstream effectors. However, in metazoa, most solid structures can be efficiently recognized. The vast physical and chemical variations of these particles exclude the possibility of preexisting receptors encoded in the host genome. This is particularly evident in the phagocytosis of materials produced after the industrial revolution, as it would not have allowed the co-evolution of any potential receptor–ligand pairs. Evidence exists that this receptor-independent solid structure uptake is also functional in modern phagocytes[8].

Binding of a rigid structure creates tension on the plasma membrane[9]. Such a tension is by itself a "sensing" mechanism that regulates coordination of membrane traffic and actin cytoskeleton[9]. Evidence suggests that membrane lipids are actively involved in phagocytosis[10]. In particular, PIP2 tends to aggregate under proper ionic environment and cytoskeletal attachment[11,12]. In addition, PIP2 is sensitive to the membrane curvature associated with lipid rafts[13,14]. However, whether PIP2 distribution itself can autonomously induce receptor-independent phagocytosis is not known.

We proposed several years ago that solid particle binding to a phagocyte surface triggers membrane lipid sorting[15,16]. This lipid redistribution leads to a strong phagocytic programing that resembles FcR-based signaling. In this report, we found that in phagocytes, an atypical ITAM sequence in Moesin, a member of the conserved ERM (Ezrin-Radixin-Moesin) family[17], transduces such an activation. Particle-induced membrane deformation leads to an autonomous accumulation of PIP2 at the site of contact. Aggregated PIP2 binds to the FERM domain of Moesin, and the latter attracts Syk to the plasma membrane via the ITAM motif. The homologs of Moesin from *Caenorhabditis elegans*, *Drosophila melanogaster*, and *Danio rerio* can also signal in place of mouse Moesin to mediate the uptake. Phylogenic analysis traces this modality of signaling to about 0.8 billion years ago, earlier than the dawn of immune receptors. This work therefore describes a general model of solid structure phagocytosis and implies a preexisting signaling platform that serves as the basis for the later development of immune tyrosine-based immune receptor signaling.

## Results

**Moesin mediates receptor-independent phagocytosis.** In the absence of receptors, crystalline structure-triggered phagocytosis is dependent on Syk[15,16], implicating the involvement of a cryptic ITAM sequence. For FcRs, the ITAM motif is either intrinsic to the receptor (such as FcRIIA) or present in the common γ chain[18] that serves to mediate Syk membrane recruitment[19]. To search for the surrogate ITAM, we used a loosely defined sequence Tyr-X-X-(Leu/Ile)-X(6-12)-Tyr-X-X-(Leu/Ile) as the probe to search the online database of PROSITE (http://prosite.expasy.org/). This search resulted in a total of 1085 independent hits in the mouse genome (Supplementary Data 1). All these hits were used to search against several NCBI GEO RNA-seq databases (https://www.ncbi.nlm.nih.gov/geo/). Among them, a total of seven

databases from bone marrow-derived DCs (BMDCs) (three), bone marrow-derived macrophages (BMDMs) (three), and RAW264.7 cells (one) were used to identify the high expressers in these cells. The expression levels were ranked by their abundance (Supplementary Data 2). The top 25 high expressers were analyzed further and functions of the corresponding proteins were annotated with UniProt as the reference (http://www.uniprot.org/). From this analysis, 18 of the 25 hits were disqualified for reasons provided in the side notes (Table 1). We produced siRNA (Supplementary Fig. 1a, b) to knock down (KD) the expression of remaining seven genes in DC2.4 cells. Among them *Hmox1* and *Csf1r* siRNA (three versions each) showed poor knockdown efficiency (Supplementary Fig. 1b), hence not pursued further. In the remaining five genes, *Lcp1* and *Msn* KD showed visible reduction of the uptake of polystyrene (latex) beads (Fig. 1a). Since *Lcp1* encodes a cytosolic protein with no known membrane binding ability[20], we focused our attention on *Msn*, or Moesin, an evolutionarily conserved structural linker protein[17]. The ITAM domain contained in Moesin was previously reported to interact with Syk in signal transduction of adhesion molecule PSGL-1; however, its role in phagocytosis was not studied[21]. To increase the knockdown efficiency, lentivirus-based shRNA with a puromycin selection marker was produced, and the knockdown cells were cloned. The selected cells showed significantly reduced ability to internalize polystyrene beads (Fig. 1b). This was accompanied by a reduction of phagocytic binding force measured by atomic force microscopy (AFM)-based single cell force spectroscopy (AFM-SCFS) (Supplementary Fig. 1c).

Evolutionarily, Moesin and other two ERM proteins belong to Band 4.1 superfamily and share significant homology with another structural protein Merlin[22,23]. They are characterized by an N-terminal FERM domain and a C-terminal actin-binding domain flanking a linker sequence[17,24]. C-terminal is also called ERM-association domain (C-ERMAD), which folds back to cover the FERM sequence, rendering the resting ERM soluble in the cytosol. Binding to the inner plasma membrane leaflet PIP2 displaces this blockage and activates ERM proteins[17,24]. As Ezrin, Radixin, and Moesin are structurally similar and share an almost identical ITAM-like sequence (Supplementary Fig. 2a)[22,25], the lone involvement of Moesin in macrophage/DC phagocytosis was likely explained by its expression level. Indeed, Moesin expression was the highest among the family members in phagocytes (Supplementary Fig. 2b), which was confirmed by online gene expression profiling database search (Supplementary Fig. 2c). In comparison with Moesin, knockdown of Ezrin and Radixin had minimal impact on phagocytic uptake of polystyrene beads (Supplementary Fig. 2d, e). To further confirm the involvement of Moesin, we co-stained actin with the three ERM proteins separately. Actin formed an even sphere around the bound beads. In comparison with Erzin and Radixin, Moesin was strongly accumulated in the phagocytic cup, and was in tight alignment with actin (Fig. 1c). Considering the potential cross-reactivities among these three antibodies[6], it was possible that Moesin antibody used in Fig. 1c inadvertently stained other ERM members. We performed the Moesin colocalization assay under Erzin and Radixin shRNA knockdown (Supplementary Fig. 3a; the knockdown efficiency and antibody specificity analysis are shown in Supplementary Fig. 3b), and the accumulation of Moesin showed no noticeable difference compared to wild-type cells. Coupled with the phagocytosis data (Supplementary Fig. 2d, e), our results suggested that Moesin was likely involved in the receptor-independent phagocytosis.

**Moesin ITAM signals in receptor-independent phagocytosis.** To ascertain the role of FERM domain ITAM as a self-sufficient signaling moiety in phagocytosis, we rescued the Moesin KD

**Table 1 Top 25 hits of highly expressed ITAM-like sequence-containing proteins in the mouse genome**

| # | Score | Gene Symbol | ITAM (-like sequence) | X(n) | Note |
|---|---|---|---|---|---|
| 1 | 56.16 | Itm2b | | | ITAM-like sequence in extracellular/lumenal domain |
| 2 | 38.70 | Fcer1g* | YTGLNTRSQETYETL | 7 | |
| 3 | 32.01 | Tyrobp* | YQELQGQRPEVYSDL | 7 | |
| 4 | 24.71 | Npc2 | | | Secretory protein |
| 5 | 22.33 | Lcp1 | YSDLSDALVIFQLYEKI | 9 | |
| 6 | 15.16 | Rps3 | | | 40S ribosomal protein |
| 7 | 9.63 | Slc6a6 | | | ITAM-like sequence in transmembrane domain |
| 8 | 9.63 | Hmox1 | YTALEEEIERNKQNPVYAPL | 12 | |
| 9 | 8.61 | Csf1r | YGILLWEIFSLGLNPYPGI | 11 | |
| 10 | 8.51 | Cd36 | | | ITAM-like sequence in extracellular domain |
| 11 | 7.96 | Atp1a1 | | | ITAM-like sequence in extracellular/transmembrane domain |
| 12 | 7.94 | Ndrg1 | YHDIGMNHKTCYNPL | 7 | |
| 13 | 6.59 | Slc11a1 | | | ITAM-like sequence in extracellular/transmembrane domain |
| 14 | 6.49 | Msn* | YLKIAQDLEMYGVNYFSI | 10 | |
| 15 | 6.10 | Myh9 | YSGLFCVVINPYKNL | 7 | ITAM-like sequence in Myosin motor domain |
| 16 | 5.86 | Actr2 | YKHIVLSGGSTMYPGL | 8 | ITAM-like sequence at ATP binding site |
| 17 | 5.42 | Rpl7 | | | 60S ribosomal protein |
| 18 | 5.15 | Iqgap1 | YSDLVTLTKPVIYISI | 8 | ITAM-like sequence at Cdc42 binding region |
| 19 | 4.96 | Ctsc | YWIIKNSWGSNWGESGYFRI | 12 | Lysosome localization |
| 20 | 4.76 | Rps25 | | | 40S ribosomal protein |
| 21 | 4.39 | Cd81 | | | ITAM-like sequence in extracellular/transmembrane domain |
| 22 | 4.14 | Degs1 | YMFLKGHETYSYYGPL | 8 | Mitochondrial localization |
| 23 | 4.09 | Glud1 | | | Mitochondrial localization |
| 24 | 3.87 | Rpl7a | | | 60S ribosomal protein |
| 25 | 3.86 | Atp6v1e1 | YQVLLDGLVLQGLYQLL | 9 | Subunit of Vacuolar ATPase |

For those analyzed further or of strong relevance to phagocytosis, their ITAM sequences are listed. The notes explain the unlikelihood of concerned sequences to serve as membrane proximal signal transducer. The gray font indicates spatial hindrances of their ITAM signaling. The proteins with functional ITAMs have been reported were indicated by "*". Red letters indicate essential amino acid residues for ITAM motif

DC2.4 cells with the full length, an ITAM-dead (Y191F + Y205F) full-length Moesin, as well as the FERM domain and ITAM-dead FERM domain constructs (Fig. 1d). Interestingly, the expression of full length or the FERM domain-only construct was sufficient to restore phagocytosis, while ITAM-dead versions failed to rescue (Fig. 1e). This result suggested that the ITAM in FERM is a signaling transducer for non-receptor-mediated phagocytosis while the structural linker function of Moesin is dispensable. Knockdown of Syk via siRNA showed a similar level of reduction in phagocytosis compared to Moesin knockdown DC2.4 cells (Fig. 2a). In the presence of MSU, while the overall Syk and Moesin levels remained the same, there was a strong association between these two molecules as revealed by Moesin co-immunoprecipitation (Fig. 2b). On the other hand, transfected Syk immunoprecipitated via Flag showed increased phosphorylation over time (Fig. 2c), and this was accompanied by increased phosphorylation of Erk1/2 and p38 (Fig. 2d). Following MSU treatment, Syk phosphorylation in Cos-1 was also increased, suggesting Syk was also involved in phagocytosis in these cells (Fig. 2e). To reveal the Moesin domain responsible for Syk binding, we produced a series of Flag-tagged Moesin truncations (Fig. 2f). In Cos-1 cells, each transcript reached similar and reproducible levels of protein expression (in molarity). Here, only the FERM domain of Moesin was able to co-immunoprecipitate Syk, while the linker and C-terminal sequences failed to do so (Fig. 2g). This result indicated that FERM of Moesin could bind to Syk, which was reminiscent of the latter's involvement in a typical FcR-mediated phagocytosis. Whether this binding correlated with the functional role of Syk in phagocytosis remained a topic for investigation.

It is important to note that in our system Moesin and Syk knockdown did not completely block phagocytosis. While this may indicate the incomplete silencing of gene expression whereby residual proteins mediated the remaining phagocytosis, it is more likely that other phagocytic mechanisms, such as integrin and complement-based cellular uptakes, were operational in addition to Moesin signaling[26,27]. We did not pursue that topic further in this report.

**Moesin signaling is downstream of PIP2 sorting.** ERM can be recruited to the plasma membrane via clustered PIP2 (refs. [28,29]). This binding induces the conformational opening of ERM[17,24,29]. Previous work has revealed that the ITAM motif of Moesin, specifically Y191 and Y205, is localized within the FERM domain of moesin between F2 and F3 subdomains, likely masked by the C-ERMAD in the closed conformation[30,31]. In contrast, PIP2-binding residues K63 and K278 in the F1 and F2 subdomains, respectively, form a cleft termed the "POCKET" in autoinhibited Moesin. Moreover, K253/K254 and K262/263 residing in the F3 subdomain formed a "PATCH" that can bind to PIP2. This PATCH is more accessible than the POCKET in the autoinhibited Moesin. It was proposed that PIP2 binds and activates Moesin progressively. First, PIP2 transiently binds to the more accessible PATCH and initiates the release to FLAP. Next, the same PIP2 molecule stably binds to the now accessible POCKET to complete conformational activation[32]. Presumably, this can lead to the

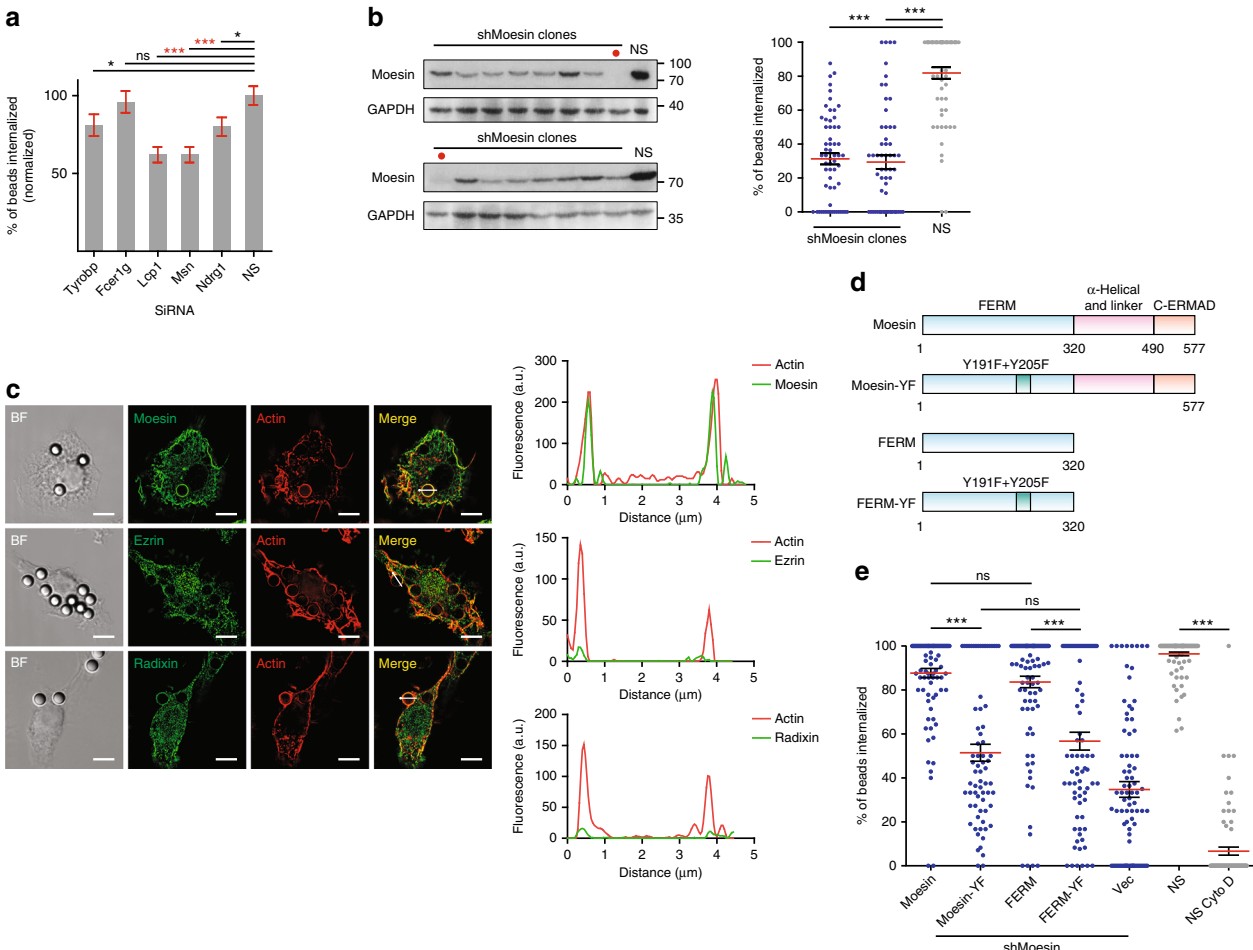

**Fig. 1** Moesin mediates receptor-independent phagocytosis. **a** Phagocytosis efficiency of DC2.4 cells transfected with non-specific (NS) or indicated siRNA. Data were presented as mean ± s.e.m. (for all following figures, unless noted otherwise), $n = 200$, collected from a total $N = 4$ independent experiments. Henceforth, "$n$" designates number of data points in each group, and "$N$" independent repeats of experiments. For indicated comparisons, ***$p < 0.001$, *$<0.05$ and ns $p > 0.05$ by one-way ANOVA with Scheffé post hoc. **b** Non-specific (NS) or Moesin shRNA-transfected DC2.4 cell single clones were immunoblotted with antibodies against Moesin and GAPDH (upper). Phagocytosis efficiency of the clones with high KD efficiency (indicated by red dot in the upper panel) is shown (lower). $n = 50$, $N = 3$. ***$p < 0.001$ by one-way ANOVA with Scheffé post hoc. **c** Left: Moesin, Ezrin, and Radixin (green) were visualized with antibodies along with actin cytoskeleton (phalloidin) on RAW264.7 cells incubated with 3 μm polystyrene beads. Structured Illumination Microscopy (SIM) was performed to obtain the fluorescence images. Scale bars, 5 μm. Right: Line profiles corresponding to fluorescence intensities of the respective ERM molecules and actin on an imaginary line across the indicated phagocytic cups. $N = 6$. **d** Schematic of Moesin and truncated Moesin fragments containing only FERM domain, or ITAM-YF mutations (Y191F + Y205F). **e** Moesin KD DC2.4 cells were rescued with the constructs in **d** by transient overexpression. Actin polymerization inhibitor Cytochalasin D was used as a negative control to establish baseline for phagocytosis. $n \geq 71$ for each group. $N = 3$. For comparison between NS and NS CytoD group, ***$p < 0.001$ by Student's $t$-test. For all other comparisons, ***$p < 0.001$ and ns $p > 0.05$ by one-way ANOVA with Scheffé post hoc

exposure of ITAM for activation and binding. In Fig. 2g, without any membrane event, the full-length Moesin did not co-precipitate Syk, suggesting a self-inhibitory state. We wondered if the Moesin FERM domain adhesion to PIP2 was sufficient to trigger its ITAM motif signaling. By immunofluorescence microscopy, line-profile analysis, and three-dimensional (3D) reconstruction, we first confirmed that in RAW264.7 cell (chosen for their better tolerance of lipid dyes) phagocytosing polystyrene beads, phosphor-ERM and PIP2 (PH-PLCδ-GFP) both showed strong colocalization with the actin-rich phagocytic cup and pERM/actin were in a close proximity (Fig. 3a, b). In Fig. 3b, pERM (T558) antibody was used as a surrogate measure for Moesin activation as it was likely the dominant ERM member in this setting. 3D reconstitution revealed a well-defined cup of Moesin conforming to a bead with actin forming an outer shell (Supplementary Movie 1). This result implies that Moesin was

indeed in its extended conformation in the receptor-independent phagocytosis. To ascertain that PIP2 was uniquely gathered during phagocytosis, phosphatidylcholine (PC), phosphatidylethanolamine (PE), and PIP2 distribution with reference to polystyrene beads were analyzed. Figure 3c shows that only PIP2 was preferentially gathered at the interface of bead/membrane interaction, while PE and PC failed to show any increased accumulation. An AFM tip was used to deliver a polystyrene bead to be in contact with the cell (Fig. 3d, scheme). A kymograph analysis showed that the accumulation of Moesin to a large extent traced the intensity change of PIP2, suggesting that PIP2 induced membrane recruitment of Moesin (Fig. 3e, and Supplementary Fig. 4a). Indeed, functional blockage of PIP2 with Geneticin (G418)[33] caused diffusion of PIP2 probe PH-PLCδ-GFP (Supplementary Movie 2 and Supplementary Fig. 4b) and inhibited the bead uptake (Fig. 3f).

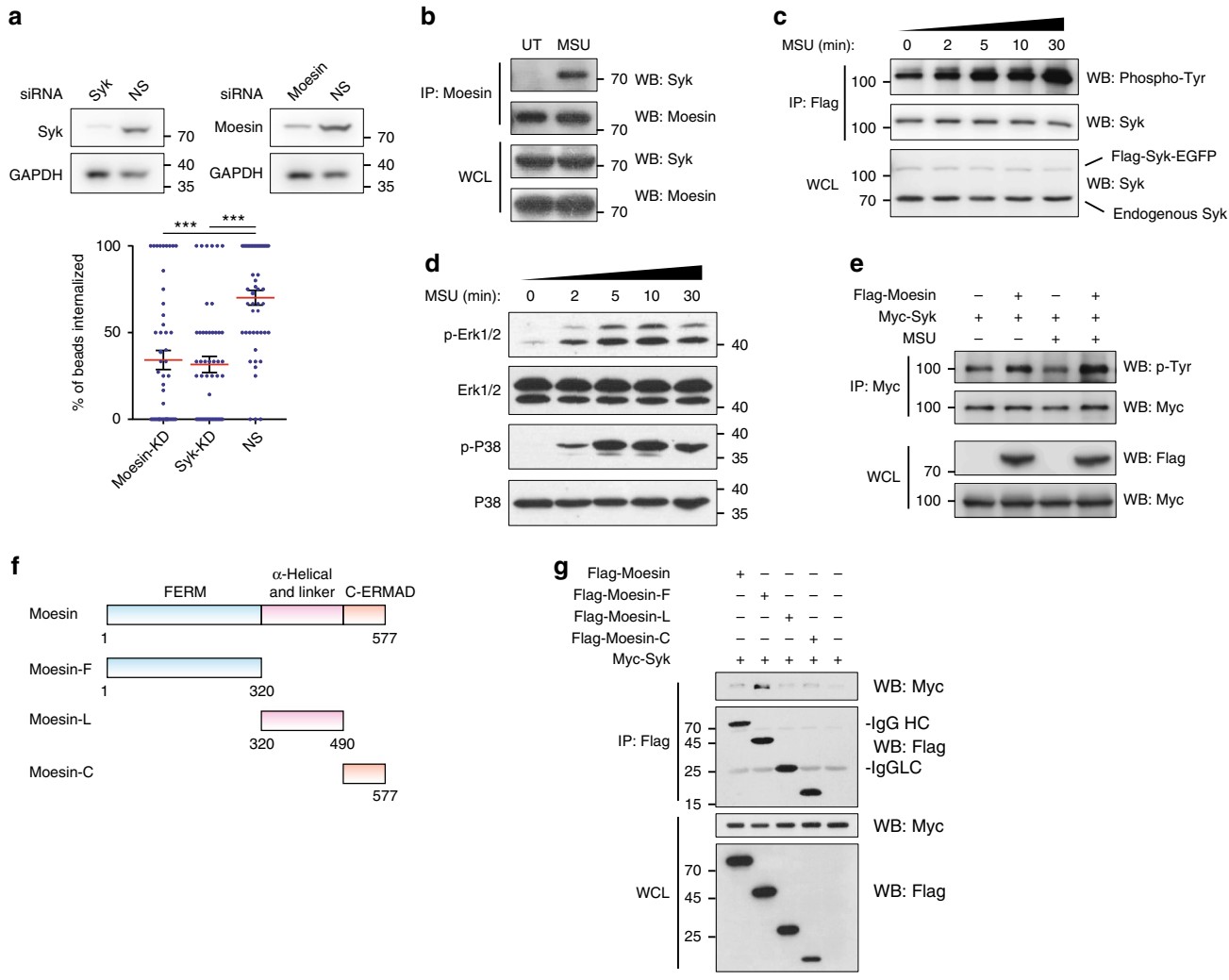

**Fig. 2** Moesin ITAM signals in receptor-independent phagocytosis. **a** Syk or Moesin siRNA-transfected or NS control DC2.4 cells were immunoblotted with antibodies against corresponding proteins, and GAPDH (upper). The phagocytosis efficiency is compared to NS control (lower). $n = 50$, $N = 3$. **b** Cell lysates from DC2.4 cells were subjected to IP with anti-Moesin. Cells were treated with $1\,\mu M$ $Na_3VO_4$ for $1\,h$ before harvesting, with or without MSU ($200\,\mu g\,ml^{-1}$) was added. $N = 3$. ***$p < 0.001$ by one-way ANOVA with Scheffé post hoc. **c** Cell lysates from DC2.4 cells transfected Flag-Syk-EGFP were subjected to IP with anti-Flag M2 beads. Cells were treated with $200\,\mu g\,ml^{-1}$ MSU crystal for indicated time before harvesting. $N = 3$. **d** DC2.4 cells were treated with $200\,\mu g\,ml^{-1}$ MSU crystal for indicated time. Total cell lysates were subjected to immunoblotting with the indicated antibodies. $N = 4$. **e** Cell lysates from Cos-1 cells transfected with Myc-Syk with or without Flag-Moesin were subjected to IP with anti-Myc. Some cells were treated with $200\,\mu g\,ml^{-1}$ MSU crystal for $1\,h$ before harvesting. $N = 3$. **f** Schematics of Moesin and the three truncated Moesin fragments containing FERM domain (F), linker region (L), and C-terminal ERM-association domain (C). **g** Cell lysates from Cos-1 cells transfected with Myc-Syk and Flag-Moesin full-length and fragments were subjected to IP with anti-Flag M2 beads. Cells were treated with $1\,\mu M$ $Na_3VO_4$ for $1\,h$ before harvesting. $N = 5$

Figure 3a–f suggest that PIP2 served as the initial reaction to an external contact by a solid structure. However, if a contact-driven PIP2 accumulation was the true initial response, it should be autonomous in response to membrane ligation in the absence of biological feedback. To confirm that PIP2 distribution was not dependent on phagocytic signaling, a bead was again delivered by an AFM cantilever to the cell surface. PIP2 accumulation at the site of contact was essentially identical between phagocytic RAW264.7 cells and non-phagocytic HEK293T cells (Fig. 3g and Supplementary Fig. 4c, d). This change was again absent for PC and PE. This further confirmed that PIP2 accumulation in response to a solid structure binding was independent of phagocytic signaling apparatus.

**Autonomous PIP2 sorting in response to physical structures.** To critically establish the notion that PIP2 redistribution to a site

of physical contact was autonomous in response to particle binding, we generated giant plasma membrane vesicles (GPMV) from surface-biotinylated RAW264.7 cells that were labeled with fluorescent PIP2, PC, or PE. Several geometric patterns made by polydimethylsiloxane (PDMS), such as round, triangular, and square, were designed with a diameter of $3\,\mu m$, and spacing of $3$–$6\,\mu m$. The initial fused silica photomask carrying the designed micropatterns was first made through a photolithography process and the Si micropatterned mold was then fabricated by high-resolution photolithography and dry etching technique. PDMS micropattern array roughly $1\,\mu m$ thick was produced by replica molding and the pattern was peeled off and attached to a glass substrate (Fig. 4a, b). The PDMS membrane was coated with NeutrAvidin to create sufficient surface anchoring. The GPMV were then delivered to cover the prefabricated patterns, and the fluorescent intensity changes of lipid in contact with each pattern

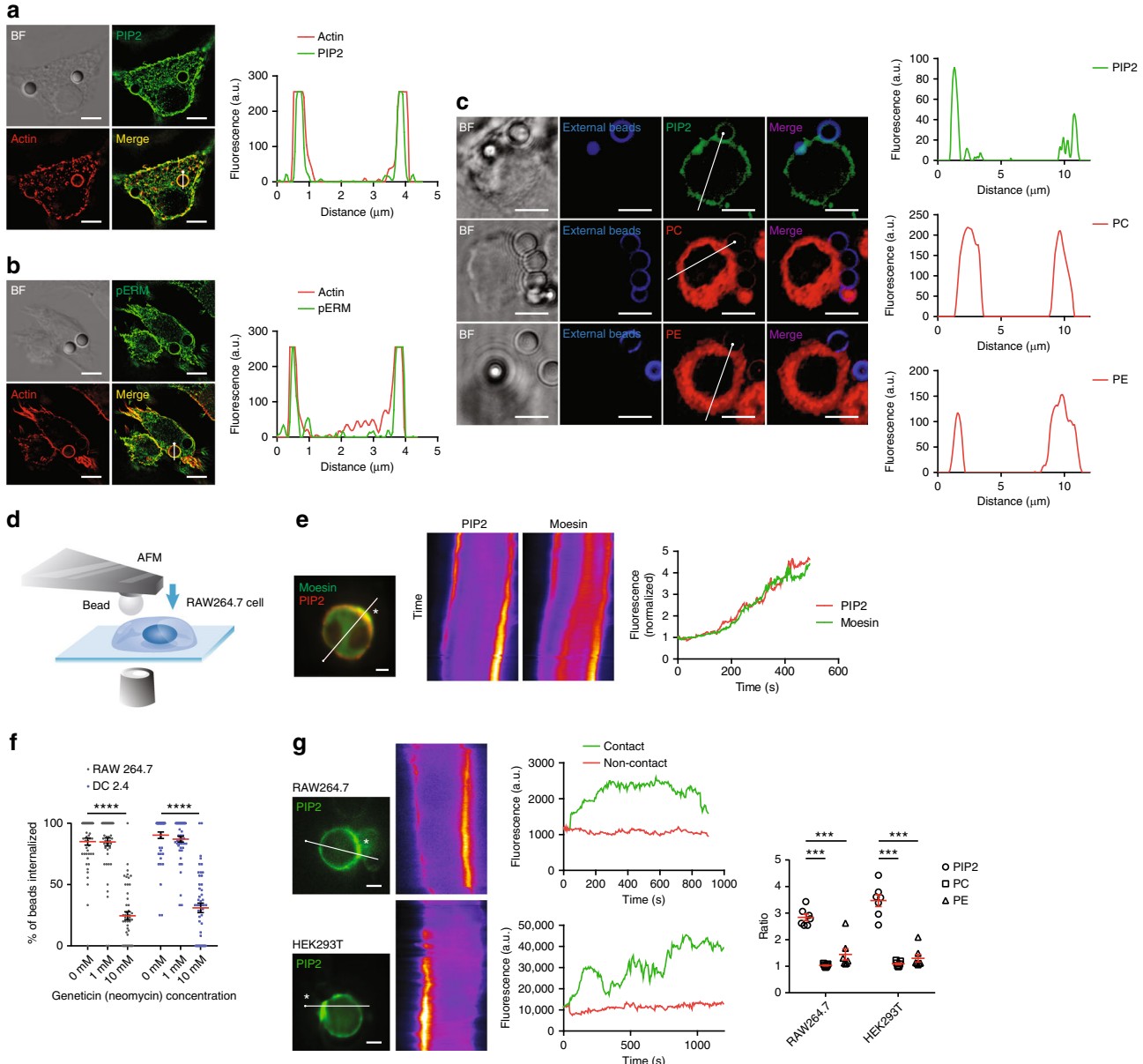

**Fig. 3** Moesin signaling is downstream of PIP2 sorting driven by solid structures. **a** PIP2 (green) was visualized with PH-PLCδ-GFP alongside actin cytoskeleton by SIM on RAW264.7 cells incubated with 3 μm naked polystyrene beads. Line profiles corresponding to fluorescence intensities of PIP2 and actin were generated across the indicated phagocytic cups. $N = 5$. Scale bars, 5 μm. **b** Identical to **a** except that pERM (T558) and actin were analyzed. Phospho-ERM were visualized with anti-pERM antibody. $N = 4$. Scale bars, 5 μm. **c** PIP2 (green), PC (red), and PE (red) were visualized with PH-PLCδ-GFP, TopFluor TMR-PC or TopFluor TMR-PE on RAW264.7 cells incubated with 3 μm biotin-BSA-coated polystyrene beads as in **a**. $N = 3$. Scale bars, 5 μm. **d** Schematic of fluorescence imaging of bead/cell contact by a bead delivered with AFM. **e** PH-PLCδ-mCherry and Moesin-EGFP were co-expressed in RAW264.7 cells. A polystyrene bead was used to contact the cell surface. Images were taken at a 6 s interval for 500 s. Localization of PIP2 and Moesin at the site of contact (indicated with "*") was examined with kymographs generated from the indicated line. The normalized fluorescence is defined as the ratio of the fluorescence intensity of PIP2 or Moesin at the site of contact over non-contact regions on the cell membrane (right). $N = 3$. Scale bar, 5 μm. **f** Phagocytosis efficiency was examined for both RAW264.7 and DC2.4 cells in the presence of 0.1 and 10 mM Geneticin. $n = 50$, $N = 4$. ****$p < 0.0001$ by one-way ANOVA with Scheffé post hoc. **g** PH-PLCδ-GFP-expressing RAW264.7 or HEK293T cells were touched with a single polystyrene bead recorded with a 6 s interval for 900 and 1200 s, respectively. Polarized distribution of PIP2 upon touching at the site of contact (indicated with "*") on cell membrane was examined with kymograph. The normalized fluorescence of PIP2, PC, and PE was calculated for RAW264.7 and HEK293T cells (right). $n \geq 10$, $N = 3$. Scale bars, 5 μm. ***$p < 0.001$ by one-way ANOVA with Scheffé post hoc

was recorded by confocal microscopy and compared with surrounding areas. Regardless of the pattern, PIP2 intensity showed ~3-fold increase in the contact regions compared to the rest (Fig. 4c). These results suggest that PIP2 accumulated in response to the contact of a physical shape without involving any cellular activities and was likely the true initiation signal of phagocytosis.

In our previous work[15,16], it was found that the MSU and alum crystal phagocytic signaling is mainly the result of ordered lipid domain (lipid rafts) aggregation driven by solid surface binding to lipid species in these domains. Work by others also indicated that PIP2 tends to colocalize with lipid rafts[24,34]. We therefore employed a ratiometric imaging method termed Generalized

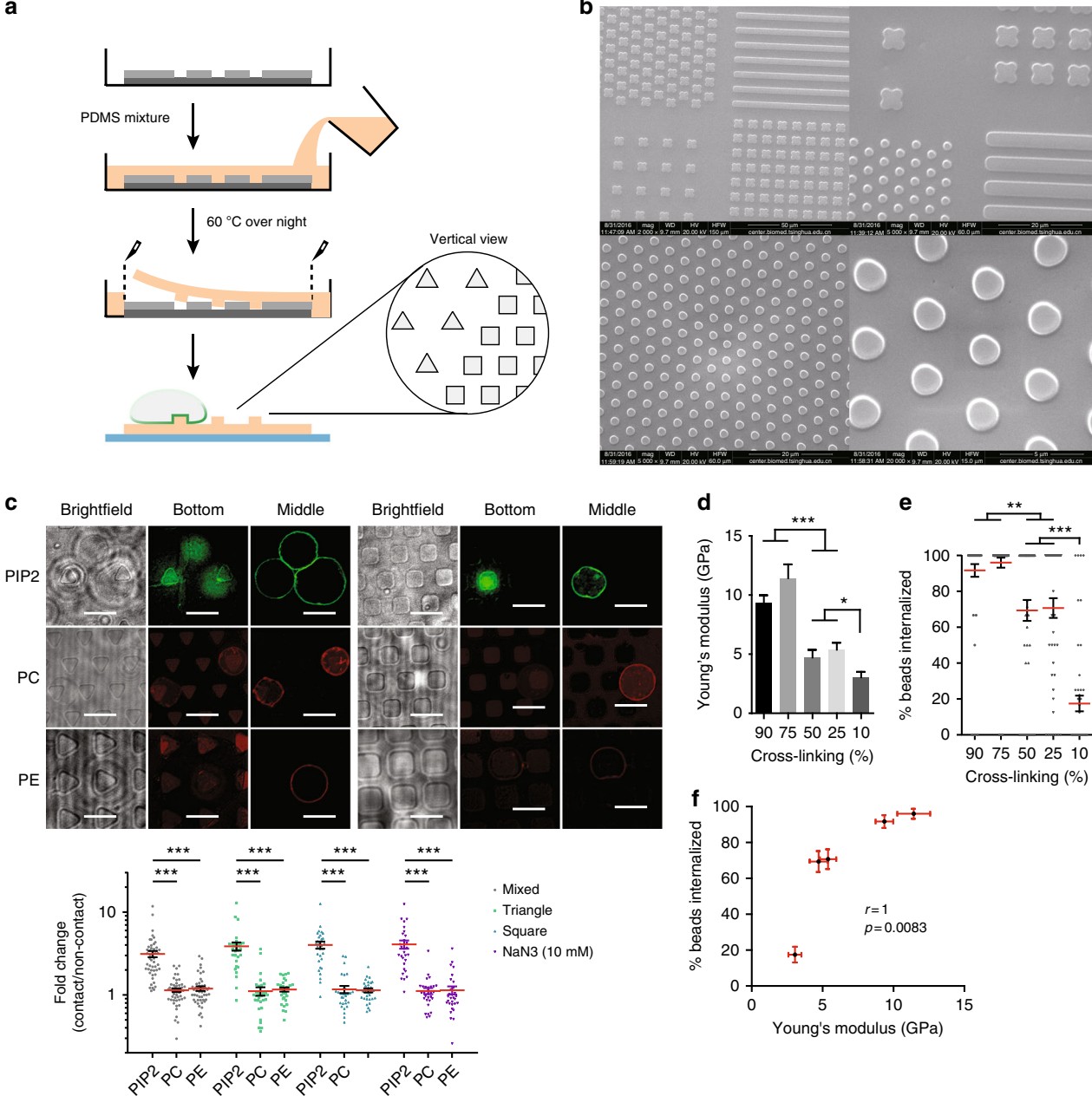

**Fig. 4** Autonomous PIP2 sorting in response to physical structures. **a** Schematic of the procedure of making PMSD pattern. **b** SEM images of PDMS micropatterns. **c** Biotinylated-GPMVs were labeled with BODIPY FL-PIP2, TopFluor-TMR PC, or PE and incubated with PDMS triangular and rectangular patterns coated with NeutrAvidin. Images were taken when GPMV were settled onto a specific pattern. Fold changes, defined as the ratio of fluorescence intensity of a given lipid at the pattern contacting regions over non-contacting regions, were measured for all shapes with or without 10 mM NaN₃. $n \geq 30$, $N = 5$. Scale bars, 5 μm. ***$p < 0.001$ by one-way ANOVA with Scheffé post hoc. **d** Polystyrene beads with different degrees of crosslinking on glass surfaces were subjected to force-indentation displacement probing with a stiff probe. The values of stiffness (Young's modulus) were obtained via indentation analysis with Veeco software. $n \geq 12$, $N = 3$. ***$p < 0.001$ and *$p < 0.05$ by one-way ANOVA with Scheffé post hoc. **e** RAW264.7 cells were incubated with 2 μm, biotin-BSA-labeled polystyrene beads with varied degrees of crosslinking for 90 min at 37 °C. The efficiency of phagocytosis was analyzed as per phagocytosis assay. $n = 50$, $N = 4$. ***$p < 0.001$ and **$p < 0.01$ by one-way ANOVA with Scheffé post hoc. **f** Correlation analysis of stiffness as shown in **d** and phagocytosis efficiency as shown in **e** by non-parametric method. Spearman's rank correlation coefficient $r = 1$. $p = 0.0083$

Polarization with c-Laurdan (a lipid-packing density sensor used to indicate lipid ordered domains)[35] to determine membrane orders as an indirect measurement for lipid rafts and investigate whether PIP2 was truly localized within the highly ordered membrane (Supplementary Fig. 4e). The results showed that, when induced by solid structures, PIP2 preferentially sorted into highly ordered membranes on RAW264.7 cells compared to control lipids (Supplementary Fig. 4e). This suggests a possible

internal link from membrane lipid binding to local PIP2 aggregation, bridged by lipid rafts.

Although two previous reports suggest that geometric shapes of phagocytic target with identical size affected the uptake[36,37], we failed to detect any difference in distinct geometry to attract PIP2. To attribute the ability of phagocytosis to a physical property of the target, we wondered if the "stiffness", a surface property extensively implicated in cellular sensing[38], was involved. A series

of polystyrene beads with variable stiffness were produced through manipulating the degree of crosslinking (from 10 to 90%), and their stiffness was determined individually by AFM based-nanoindentation (Fig. 4d). As a trend, the degree of crosslinking was as expected associated with the stiffness, which was echoed by the degree of phagocytosis (Fig. 4e). Non-parametric correlation analysis resulted in a Spearman coefficient $r = 1$ ($p = 0.0083$), indicating that stiffness of phagocytic target is positively correlated with phagocytic efficiency (Fig. 2f). This analysis suggests that in addition to the contour of a solid structure, there may be a cut off in "hardness"; objects with a surface value below this limit are inefficient in triggering phagocytosis.

**PIP2-moesin-syk axis vs. fcr signaling**. Since particle binding is sufficient to drive PIP2 sorting, we wondered if any membrane ligation with sufficient force can trigger Moesin-based phagocytosis. Complete CD4 or chimeric extracellular/transmembrane CD4/intracellular Moesin ITAM construct was transfected into Cos-1, Cos-7, HEK293T, and NIH3T3 cells, along with an ITAM-dead control (Fig. 5a). As expected, in mildly phagocytosing Cos-1 cells, polystyrene beads coated with anti-CD4 IgG were efficiently phagocytosed, similar to FcRIIA-transfected cells in phagocytosing antibody-opsonized beads (Fig. 5b). Minimally phagocytosing Cos-7 cells showed relatively lower phagocytosis; however, the "mock" CD4 receptor and FcRIIA demonstrated similar efficiencies (Fig. 5b). NIH3T3 (fibroblast) and HKE293T (embryonic kidney cells), however, were mostly unresponsive. Notably, in Cos-1 and Cos-7 cells, all three versions of CD4 phagocytosed anti-CD4 antibody-coated beads, suggesting that in the presence of endogenous Moesin, surface ligation itself was sufficient. In contrast, in Moesin KD cells, only CD4 with an intact Moesin ITAM mediated phagocytosis with high efficiency, in comparison with CD4 alone or with the ITAM-dead mutant (Fig. 5c). This result suggests a scenario that sufficient binding force exerted on the membrane from a solid particle can induce phagocytosis via Moesin; in the absence of Moesin, the ITAM built into the "pseudo" receptor can provide similar stimulatory signal for phagocytosis, reminiscent of a phagocytic receptor. These results collectively indicate that Moesin ITAM domain can signal following surface ligation or behave as a signaling motif in a "receptor"-like structure. Importantly, this "pseudo" receptor signaling was similar to FcR-based phagocytosis in its sensitivity to PI3K, Syk, Cdc42, and Src inhibitors (Supplementary Fig. 5a), suggesting a signaling cascade similar to the typical FcR-mediated phagocytosis. Of note, the "pseudo" receptor signaling on DC2.4 cells displayed the same sensitivity (Supplementary Fig. 5a), suggesting again a shared signaling cascade.

In the "pseudo" receptor analysis, there was a potential confounding factor in its signaling cascade. Src family kinases are important for ITAM phosphorylation. In line with this notion, Src inhibitors AZD0530 and PP2 reduced the phagocytosis (Supplementary Fig. 5a). As CD4 is known to recruit Lck[39], whether the latter, although typically restricted to T cells[40], was involved was not clear. To this end, we expanded the analysis. We expressed mouse ICAM-1 and CD8α on Cos-1 cells and successfully induced significant levels of phagocytosis of anti-ICAM-1 and anti-CD8α-coated latex beads (Supplementary Fig. 5b, c). Similarly, whether these molecules carried Moesin ITAM did not significantly impact the uptake efficiency. Under Moesin knockdown, however, the chimeric ITAM sequence became critical. (Supplementary Fig. 5b, c). These additional analyses have two implications. First, it is likely that other Src family members may work in place of Lck as ICAM-1 was not known to engage this particular kinase. Second, Moesin ITAM's

signal capacity to induce phagocytosis in "pseudo" receptors is likely a generalizable effect, not limited to CD4 molecule.

Although Syk binding to FcR-associated ITAM is similar to its binding to Moesin, we wondered if the former represents an evolutionarily more advanced/efficient form of phagocytic signaling. First, in comparison with the standard FcR signaling, we noticed that Moesin-mediated Akt and Erk activation was nearly identical (Supplementary Fig. 6a), suggesting both mechanisms were activation-wise similar. In Moesin KD cells, IgG-coated beads were phagocytosed as efficiently as WT cells, suggesting that FcRs no longer require Moesin (Fig. 5d). Without IgG coating, the phagocytic efficiency was reduced in the absence of Moesin (Fig. 5d). In the phagocytosis of 1, 3, and 6 μm beads, IgG-coated ones in comparison showed high efficiency in taking up smaller beads, and had better capacity to internalize the larger ones (Fig. 5e). A similar analysis indicated that IgG coating promoted small bead internalization in terms of the number of uptakes per cell (Fig. 5f). With large sizes, the overall volume and surface area internalized were increased in the presence of IgG coating (Fig. 5g). Therefore, FcRs appear to have evolved to become independent of Moesin and demonstrate a much better efficiency in solid particle phagocytosis.

In contrast to FcR-based phagocytosis, there are several phagocytic mechanisms that do not seem to involve an ITAM-like structure. Scavenger receptors are known as non-opsonic phagocytic receptors for the lack of any apparent signaling motif in their intracellular domains[41]. They are believed to engage their targets via polycationic and polyanionic interactions (the electrostatic patch model)[42]. To test whether these receptors can also use Moesin as the conduit to relay the extracellular binding across the membrane, polystyrene beads were coated with MalBSA (maleylated-BSA, a ligand for SR-A6/SR-A1, previously known as MARCO/SR-A)[21]. This treatment increased the phagocytosis in comparison to BSA alone-coated beads (Supplementary Fig. 6b); the effect was mediated via SR-A6/SR-A1 as macrophages from mice with deficiencies in these two receptors did not show enhanced phagocytosis of MalBSA-coated targets, in comparison with similar macrophages from WT mice. However, in the absence of Moesin, MalBSA did not show any enhanced phagocytosis over BSA alone (Supplementary Fig. 6c), suggesting that Moesin was involved in the SR-A6/SR-A1-mediated phagocytosis.

**Evolutionary implications of PIP2 signaling in immune signaling**. Considering the possibility that Moesin-based signaling was evolutionarily conserved, we cloned the ERM protein from *Caenorhabditis elegans*, *Drosophila melanogaster*, and *Danio rerio* (Fig. 6a). Upon transfecting these sequences into Moesin KD DC2.4 cells, phagocytosis was restored, suggesting ERM proteins are conserved through the evolution as a phagocytic signaling mechanism in the absence of FcRs (Fig. 6b).

To identify the origin of ERM signaling, phylogenic analyses based on model organisms were performed (see Methods for technical details). For Moesin, the evolutionary lineages furthest to *H. sapiens* found by BLAST were *D. melanogaster* and *C. elegans*, which diverged from the vertebrates at about 797 million years ago, indicating the latest time of hypothetic origin of Moesin (Fig. 7; Table 2 as the amino acid substitution table; Supplementary Figs. 7, 8a). The search for Syk traced back to a common origin of 435 million years ago. However, the times of common ancestors of the PI3K catalytic subunit, and of Syk and its analogous ZAP70 were found as early as 1.496 and 1.032–0.757 billion years ago, respectively (Fig. 7; Supplementary Figs. 7, 8b, c). The earliest identifiable origin of ITAM-containing surface molecules such as Fc common γ chain, CD3 ζ chain, and

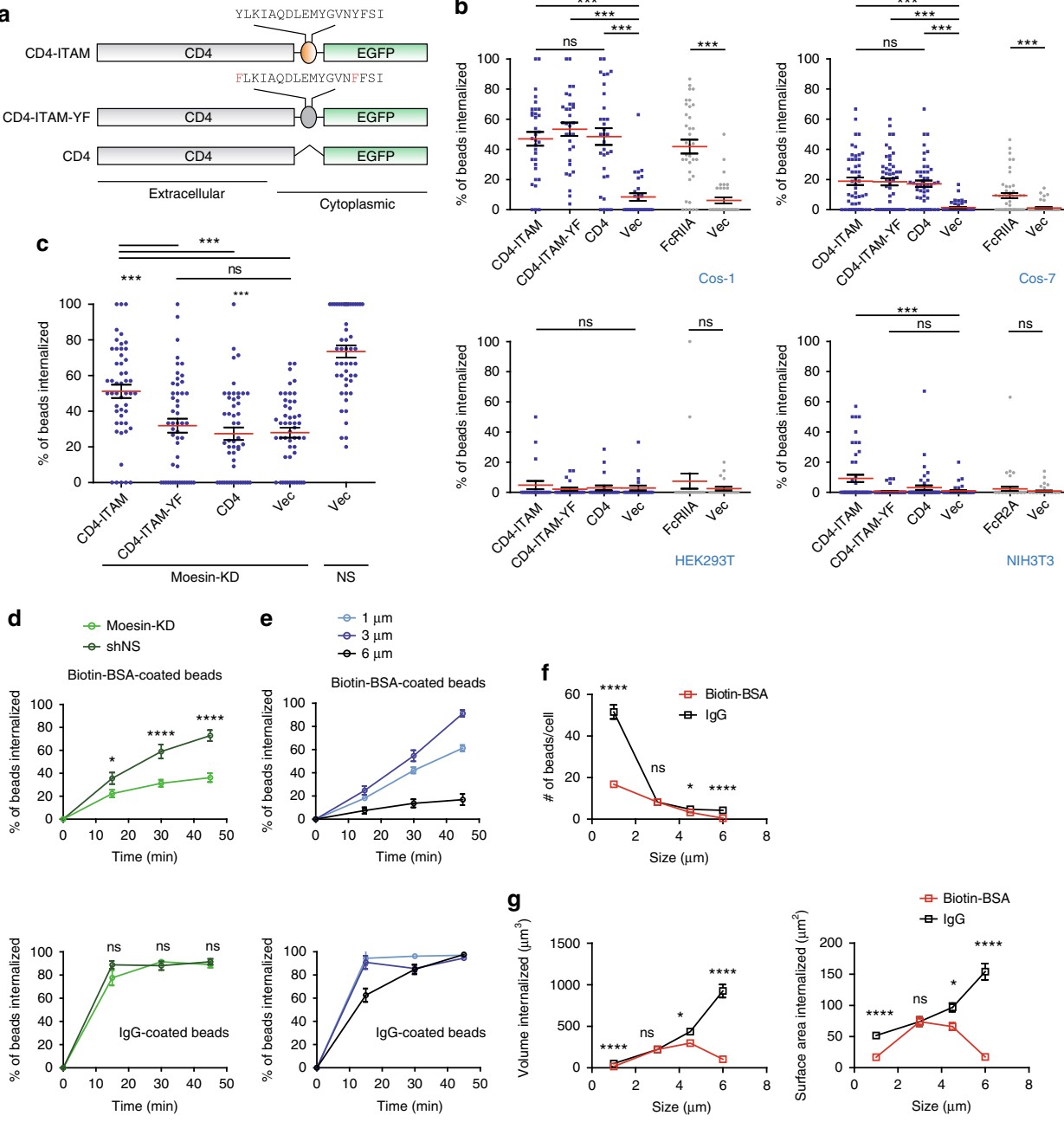

**Fig. 5** PIP2-Moesin-Syk axis is qualitatively similar to but quantitatively less robust than FcR signaling. **a** Schematic of a series of CD4-EGFP chimeric sequences, with or without a WT or mutated ITAM. **b** Cos-1, Cos-7, HEK293T, and NIH3T3 cells were transfected with the chimeric molecules by transient overexpression. Empty vector control and FcRIIA for phagocytosis efficiency control were also shown. ***$p < 0.001$ and ns $p > 0.05$ by one-way ANOVA with Scheffé post hoc. Phagocytosis efficiency was measured with anti-CD4-coated or IgG-coated 3 μm beads. $n = 50$, $N = 3$. **c** Identical to **b** except that Moesin KD DC2.4 cells were rescued with the constructs as shown in **b**. NS was used as a positive control to establish the full capacity of phagocytosis in DC2.4 cells without Moesin knockdown. $n = 50$, $N = 3$. ***$p < 0.001$ and ns $p > 0.05$ by one-way ANOVA with Scheffé post hoc. **d** Efficiency of Moesin-dependent (top) and FcR-dependent (bottom) phagocytosis was measured with biotin-BSA-coated beads or IgG-coated beads in shMoesin or shNS treated DC2.4 cells. $n = 50$, $N = 3$. *$p < 0.5$, ****$p < 0.0011$ and ns $p > 0.05$ by one-way ANOVA with Scheffé post hoc. **e** Impact of particle size on the efficiency of Moesin-dependent and FcR-dependent phagocytosis was measured as per phagocytosis assay with 1, 3, or 6 μm biotin-BSA-coated beads (upper) or IgG-coated beads (lower) in RAW264.7 cells. $n = 50$, $N = 4$. **f** To investigate the effects of particle sizes on phagocytosis, phagocytosis assays were performed with 1, 3, 4.5, and 6 μm biotin-BSA-coated (red line) or IgG-coated (black line) polystyrene beads on RAW264.7 cells. The numbers of beads internalized per cell on average were counted for each condition. $n = 50$, $N = 4$. *$p < 0.05$, ****$p < 0.0001$ and ns $p > 0.05$ by one-way ANOVA with Scheffé post hoc. **g** Volume and surface area internalized by RAW264.7 cells in **f** were calculated from the following formula. Volume $= 4/3\pi r^3$, and surface area $= 4\pi r^2$. *$p < 0.5$, ****$p < 0.0011$, and ns $p > 0.05$ by one-way ANOVA with Scheffé post hoc

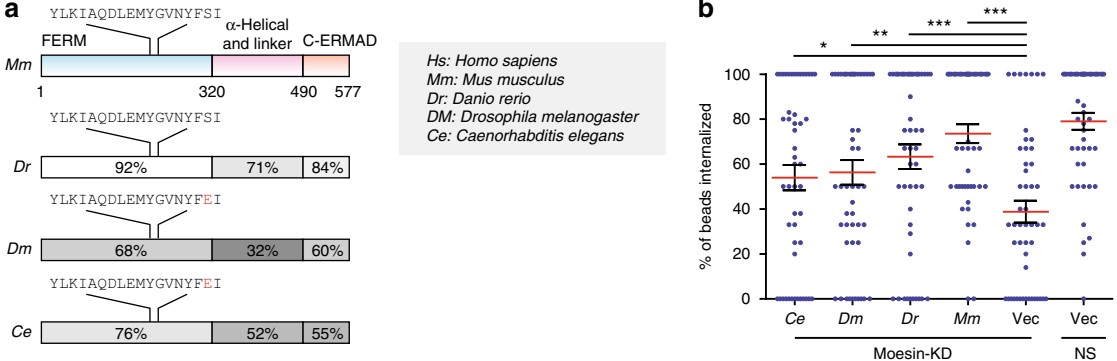

**Fig. 6** Moesin function is conserved in solid structure phagocytosis. **a** Schematic of Moesin proteins in *Mus musculus (Mm)*, *Danio rerio (Dr)*, *Drosophila melanogaster (Dm)*, and *Caenorhabditis elegans (Ce)*, identified by NCBI Blast search. Functional domains and sequences of ITAM motif are shown. Two-letter abbreviations of all the listed species presented in this figure are shown on the left. **b** Moesin KD DC2.4 cells were rescued with the molecules indicated in **a** by transient overexpression. Vector transfection of the KD and WT cells were also shown. $n = 50$, $N = 3$. *$p < 0.05$, **$p < 0.01$, and ***$p < 0.001$ by one-way ANOVA with Scheffé post hoc

DAP12 by BLAST was also around 435 million years ago, although its function is not clear (Fig. 7; Supplementary Figs. 7, 9a–c). The RAG transposon invasion into the immunoglobulin superfamily of the vertebrate genome took place about 600 million years ago[43], and the vertebrate genome duplications were estimated to be also around 600–700 million years ago (Fig. 7; Supplementary Fig. 7). Those events provided the "tools" and the "materials" for the emergence of modern immune receptors[44,45]. The high diversity of immune receptors (FcRs, BCR, and TCR)-based signaling cascade (ITAM-Syk/ZAP70-PI3K) could undergo the diversifications in specificity determinations to give rise to the modern immune receptors only after those two events. As expected, the coalescent time of the FcR II family members was only 90 million years ago, followed by a rapid radiation (Fig. 7; Supplementary Figs. 7, 9d). Therefore, modern FcR-based phagocytosis that relies on the Fc portion of antibodies cannot account for the phagocytosis beyond this time point. ERM-based phagocytosis is a likely candidate for an evolutionarily conserved form of solid particle uptake, already in the full complement of immune tyrosine-based apparatus.

## Discussion

Figure 8 and Supplementary Movie 3 suggest a model on how the primordial Moesin-based signaling was gradually adopted by the modern FcR-based phagocytosis, and how the intracellular ITAM-Syk signaling cascade is shared by two different phagocytic mechanisms.

ERM protein emerged from the founder Band 4.1 at the transition towards multicellularity[46]. The role of Moesin has been mostly recognized as a structural linker, similar to other ERM members[47]. It is believed that PIP2/ERM interaction mainly serves as an initial membrane recruitment and ERM in turn stabilize cortical actin[48,49]. In recent years, the extended functions of ERM members are increasingly appreciated[50–53]. Yet, the involvement of its ITAM is poorly understood. In one case, it was reported to signal downstream of PSGL-1 (ref. [21]). Structurally, the ITAM motif is well maintained across FcR γ chain, CD3 chains including ζ, Igα/β, and DAP12, with exceptions of Dectin 1 (YXXXL) and FcRIIA (11 AA spacer). We found that in addition to the 10 AA spacer, all ERM ITAM have a lysine following the first tyrosine, which in the case of most immune ITAM is an acidic residue. Despite these differences, our results show that Moesin ITAM contributes to an FcR-like phagocytic event. There remains the important question whether the linker and actin-

binding domains of Moesin are also involved in the overall phagocytosis. In FcR-based phagocytosis, ITAM-containing transducers (such as FcγRIIA and the common γ chain) are not known to have actin-binding capacity[54]. As membrane anchoring effects of the linkers are important, the latter is likely provided in trans. Our data seem to suggest that FERM domain, once attached to PIP2, can function in a manner similar to the canonical ITAMs. As in our experiment Moesin knockdown is not complete, the residual Moesin may again play the role of membrane anchoring along with other linkers, such as other ERM and spectrin[55]. Whether an actin-binding domain cis to the ITAM would render the phagocytosis more efficient is an interesting question for the future.

Our current understanding of PIP2 functions focuses on several fronts: precursor of messengers, attachment anchors, and membrane organization[24]. It contributes to the negative membrane potential in the inner leaflet that forms the "electrostatic switch" to help organize signaling molecules via their cationic charges[56]. PIP2 is mostly localized at and constitutes 1–2% of lipids to the inner leaflet[19]. Its distribution is a balanced outcome of its synthesis and metabolism[20,28]. However, it is important to note that PIP2 is regulated by lipid rafts and membrane curvature[24,34]. Cholesterol has an "innate" ability to regulate the richness of PIP2 (refs. [13,34,57]). Previously we reported that lipid raft aggregation is the initiation mechanism of receptor-independent phagocytosis. We speculated that since solid structures attract lipid rafts, an ITAM sequence contained in one or more raft-associated proteins may serve as the signal transmitter across the plasma membrane[16]. However, deletion of individual ITAM-containing receptors such as DAP12 and FcR γ chain did not produce an anticipated effect in reducing solid particle phagocytosis[16]. In light of this report, it is likely that aggregation of lipid rafts triggers the accumulation of PIP2, which in turn serves as the docking site of Moesin in its extended configuration, exposing the ITAM domain. In other words, Moesin ITAM and PIP2 appear to bridge the missing link in a relay order of lipid raft aggregation-PIP2 enrichment-Moesin membrane recruitment-Syk activation.

Phagocytosis is a complex process involving multiple interdependent players[8]. Nevertheless, the specificity determination by FcRs clearly implicates the "initiation" event that emanates downward to the entire intracellular signaling cascade. PIP2 is a highly dynamic molecule and serves the phagocytic signaling in a temporally precise manner. It accumulates upon the solid target binding, and rapidly dissipates from the front edge upon phagocytic cup insertion[58]. This regulation may serve the purpose of

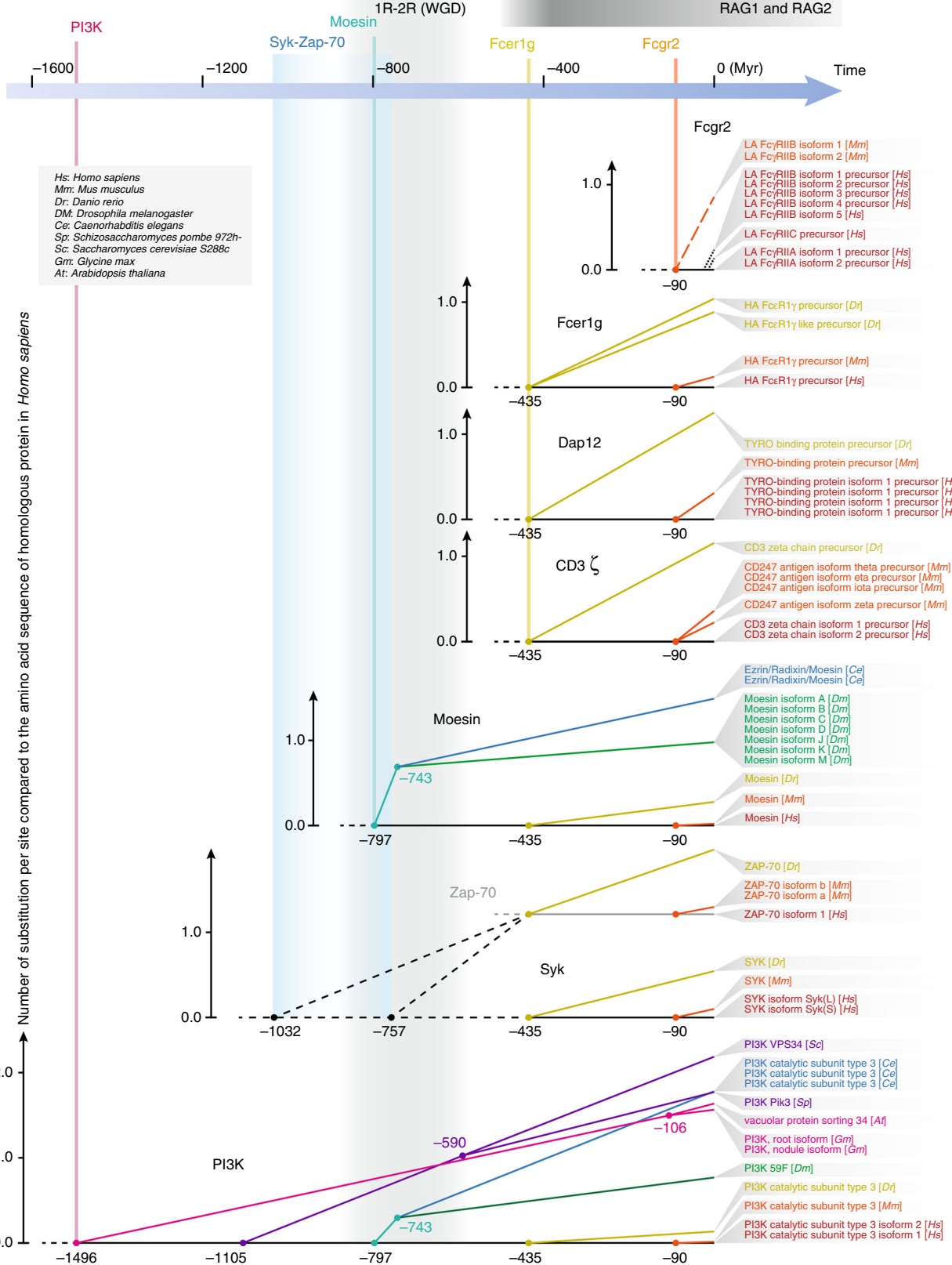

secondary lipid messenger formation as well as precise recruitment and expulsion of other signaling molecules[13,59]. To study its involvement in our system, several assays in this report were designed to identify the original "innate" response to a solid particle contact. The motion patterns of PIP2 in non-phagocytes and in GPMV's response to fabricated micropatterns imply that

the local distribution of this lipid may be the first available intracellular beacon to herald an extracellular contact. This proposal is in line with PIP2's propensity to be organized by membrane curvature and lipid domains.

Our results suggest that the PIP2/Moesin-mediated signal transduction was likely operational prior to the RAG transposon

**Fig. 7** Potential evolutionary implications of PIP2-based signaling in modern immune receptor-based phagocytosis. Phylogenetic trees of Fc γ receptor II family proteins (Fcgr2), Fc ε receptor γ subunit (Fcer1g), DAP12 (Tyrobp), CD3 ζ chain, Moesin, Syk family proteins (ZAP70 and Syk), and PI3 kinase catalytic subunit (PI3K) are shown. On top of the figure, the time axis is shown in the unit of million years. For each tree, the time axis is shown on the bottom, and is aligned to the time axis on the top. The species divergence times from published studies are used. For each tree, the vertical axis on the left shows the number of substitution per site compared to the amino acid sequence of the homolog in *Homo sapiens*, so the slope of each branch represents amino acid substitution rate. The gray column indicates the time interval of two rounds of whole-genome duplication (WGD) during the evolution of common ancestors of chordates (1R) and vertebrates (2R). The dark gray bar on top of the figure indicates the time interval of emergence of RAG1 and RAG2 (not the RAG invasion itself)

### Table 2 Amino acid substitution models selected by SMS

| Protein | Model |
|---|---|
| Moesin | LG + G + I + F |
| Phosphatidylinositol 3-kinase catalytic subunit type 3 | LG + G + I + F |
| Tyrosine-protein kinase SYK/ZAP70 | JTT + G + I |
| High-affinity immunoglobulin epsilon receptor subunit gamma | WAG + G |
| Low-affinity immunoglobulin gamma Fc region receptor II-a | JTT + G |
| TYRO protein tyrosine kinase-binding protein | JTT |
| T cell surface glycoprotein CD3 zeta chain | VT |

invasion and the 2R genome duplication[23], which certainly predated the FcR diversification. This seems to suggest that the ERM/Syk signaling platform had been in existence before the FcRs, and the immune receptor signaling "hijacked" this platform. This report provides evidence that phagocytosis can be induced by surface ligation-mediated lipid sorting, which fulfills the conceptual void how phagocytes can recognize almost inexhaustible variations of solid structures. In addition, this report proposes a potential link of how ancient, relatively inefficient form of particle uptake is linked to modern opsonization-based highly efficient form of phagocytosis, and how the conserved ITAM/Syk-based elements may have been used throughout the evolution and progressed to shoulder the lion's share of sophisticated immune signaling in modern leukocytes.

## Methods
**Mice and cells**. C57BL/6 mice were housed at Laboratory Animal Research Center of Tsinghua University. All experiments performed were approved by the animal research committee at the Tsinghua University.

DC2.4 and RAW264.7 cells were cultured in RPMI-1640 medium supplemented with 10% fetal bovine serum (FBS), HEPES, penicillin/streptomycin, and 2-ME. Cos-1, Cos-7, HEK293T, NIH3T3 cells were cultured in Dulbecco's modified Eagle's medium (DMEM) containing 10% FBS. SR-A1$^{-/-}$ SR-A6$^{-/-}$ double knockout bone marrow from C57BL/6 mice and wild-type bone marrow were grown in media containing 25% of L929 supernatant.

**Reagents**. Anti-Moesin (ab151542) (1:1000), anti-Ezrin (ab4069) (1:200), anti-Radixin (ab52495) (1:1000), anti-phospho-ERM (ab76247) (1:1000), and anti-CD8α (ab209775) were purchased from Abcam. anti-Zap-70 (2705) (1:1000), anti-Syk (2712) (1:1000), anti-phospho-Erk1/2 (8544, 9101) (1:1000), anti-Erk1/2 (9102) (1:1000), anti-phospho-Akt (9271) (1:800), anti-Akt (9272) (1:1000), anti-phospho-P38 (9211) (1:1000), anti-P38 (9212) (1:1000), anti-rabbit-IgG (7074), and anti-mouse-IgG (7076) were purchased from CST. Anti-phospho-tyrosine (05–321) (1:1000) was purchased from Merck. Anti-GAPDH (BE0023) (1:3000) was purchased from EASYBIO. Anti-Myc (M20002) (1:1000), anti-Flag (M20008) (1:1000), and Protein A/G-Agarose (A10001) were purchased from Abmart. Anti-Flag M2 affinity gel (A2220) and uric acid (U2625) were purchased from Sigma-Aldrich. Polystyrene microspheres (19822) were purchased from Polysciences. Piceatannol (527948), Syk Inhibitor (574711), Syk Inhibitor IV, BAY 61-3606 HCL (574714), R406 (5.05819.0001), Wortmannin (681676), and LY294002 (440204) were purchased from Calbiochem. AZD0530 (S1006) and PP2 (S7008) were purchased from Selleck. Bodipy FL-PI(4,5)P2 (C-45F6) were purchased from Echelon Biosciences. TopFluor TMR-PC (810180C) and TopFluor TMR-PE (810241C) were purchased from Avanti Lipids. Geneticin (10131027) was purchased from ThermoFisher.

**Plasmids**. PH-PLCδ-GFP (Addgene #35142), PH-PLCδ-mCherry (Addgene # 36075), and Lifeact-TdTomato (Addgene # 54528) vectors are purchased from Addgene. Lifeact-EGFP (60112) was purchased from Ibidi. PH-PLCδ has preferential binding to PIP2 and was therefore chosen. Full-length Moesin and truncated Moesin were constructed by cloning indicated fragments from mouse cDNA into pCAG-IRES-Neo vector[60] fused with a C-terminal EGFP. CD4-ITAM-EGFP and CD4-EGFP were constructed by cloning mouse CD4 with or without a C-terminal ITAM from Moesin, followed by an EGFP fragment. Flag-tagged full-length Moesin and truncations were constructed by cloning the fragments into pcDNA3.1 fused with an N-terminal Flag tag. 6Myc-tagged Syk was constructed by cloning mouse Syk into pCS2 with an N-terminal 6x Myc tag. All mutated forms were generated by site-directed mutagenesis and confirmed by DNA sequencing.

**ITAM screening**. An ITAM sequence Tyr-X-X-(Leu/Ile)-X(6–12)-Tyr-X-X-(Leu/Ile) was used as the probe to search the online database of PROSITE (http://prosite.expasy.org/scanprosite/). Proteins were sorted by their expression levels according to three previous RNA-Seq studies in NCBI GEO Database (GSE62704: GSM1531752, GSM1531768 for BMDC; GSE63199: GSM1543756, GSM1543757, GSM1543758, GSM1543759 for BMDM; GSE52320: GSM1263072 for RAW264.7). The top expressers of ITAM-sequence-containing proteins were analyzed further. The functions of these proteins were annotated based on UniProt (http://www.uniprot.org/).

**Transfection and RNA interference**. Plasmid DNA transfection of DC2.4 and other cell lines was performed using Lipofectamine 2000 (Invitrogen) or Neofect transfection reagent (Neofect Biotech) according to the instructions of the manufacturers. RAW264.7 cells were transfected with XFect transfection reagents (Clontech 631318) as per the manufacturers' instructions. INTERFERin siRNA transfection reagent (Polyplus) was used to transfect siRNA. SiRNA sequences were as follows, and non-specific siRNA control was purchased form Genepharma. siCsf1r#1: 5′-GCUCUUUCUGAACCGUGUAAA-3′; siCsf1r#2: 5′-CCUACUCAGUUGCCCUACAAU-3′; siCsf1r#3: 5′-CCAUGGUGAAUGGUAGGGAAU-3′; siFcer1g#1: 5′-CCUACUCUACUGUCGACUCAA-3′; siFcer1g#2: 5′-CAGCUCUGCUAUAUCCUGGAU-3′; siFcer1g#3: 5′-UGAGACUCUGAAGCAUGAGAA-3′; siHmox1#1: 5′-GCCGAGAAUGCUGAGUUCA-3′; siHmox1#2: 5′-ACAGUGGCAGUGGGGAAUUUAU-3′; siHmox1#3: 5′-AGCCACACAGCACUAUGUAAA-3′; siLcp1#1: 5′-GCUUUGAUGAGUUUAUCAA-3′; siLcp1#2: 5′-GCCAAGUAGCUUGCCUAUUAA-3′; siLcp1#3: 5′-CGAUGGCAUAGUUCUUUGUA-3′; siMsn#1: 5′-GGACGAGGACAAAUACAAGACCCUGC-3′; siMsn#2: 5′-GGAGCGUGCUCUCCUGGAA-3′; siMsn#3: 5′-CGGUCCUGUUGGCUUCUUA-3′; siNdrg1#1: 5′-AACUCAUUCCUGGAAACAAA-3′; siNdrg1#2: 5′-CCUGGAAACAAACUUCUGUUU-3′; siNdrg1#3: 5′-CCUACGCUAAAUGCGGUAUUAA-3′; siTyrobp#1: 5′-CCAAGAUGCGACUGUUCUU-3′; siTyrobp#2: 5′-GGUGUUGACUCUGCUGAUU-3′; siTyrobp#3: 5′-GGGACCCGGAAACAACACA-3′; siSyk#1: 5′-ACCUCAUCAGGGAAUAUGU-3′; siSyk#2: 5′-GACUAUGAAUGCCCACUAAAU-3′; siSyk#3: 5′-CCUCAUCAGGGAAUAUGUG-3′.

**Generation of stable cell lines**. DC2.4 cells expressing Flag-Syk-EGFP and Moesin/Ezrin/Radixin knockdown were established as follows: briefly, pCAG-ires-Neo vector harboring Flag-Syk-EGFP was delivered into DC2.4 cells with Lipofectamine 2000. After 2 weeks selection with G418 (600 μg ml$^{-1}$), clones were picked and characterized for EGFR expression. For knockdown, DC2.4 cells were infected with lentivirus packaged by Moesin/Ezrin/Radixin and negative control shRNAs, which were purchased from Sigma-Aldrich (MISSION® shRNA Library). Forty-eight hours post-infection, cells were cultured in puromycin (1 μg ml$^{-1}$) and the medium was changed every 2 days. After about 1 week, single clones were picked and verified by western blots.

**Phagocytosis assay**. Cells were seeded into 24-well plates at 10$^5$ cells per well the night before the experiment. Biotin-BSA-coated beads resuspended in Opti-MEM at 1:1000 were centrifuged to the bottom of the dish at 930 *g* for 3 min at 4 °C. Cells were chilled on ice for 5 min before being moved into a 37 °C, 5% CO$_2$ incubator. Cells were incubated with beads for indicated durations, then moved from the incubator and washed with phosphate-buffered saline (PBS) buffer once before 4% PFA fixation for 10 min. After fixation, cells were directly labeled with 1:200

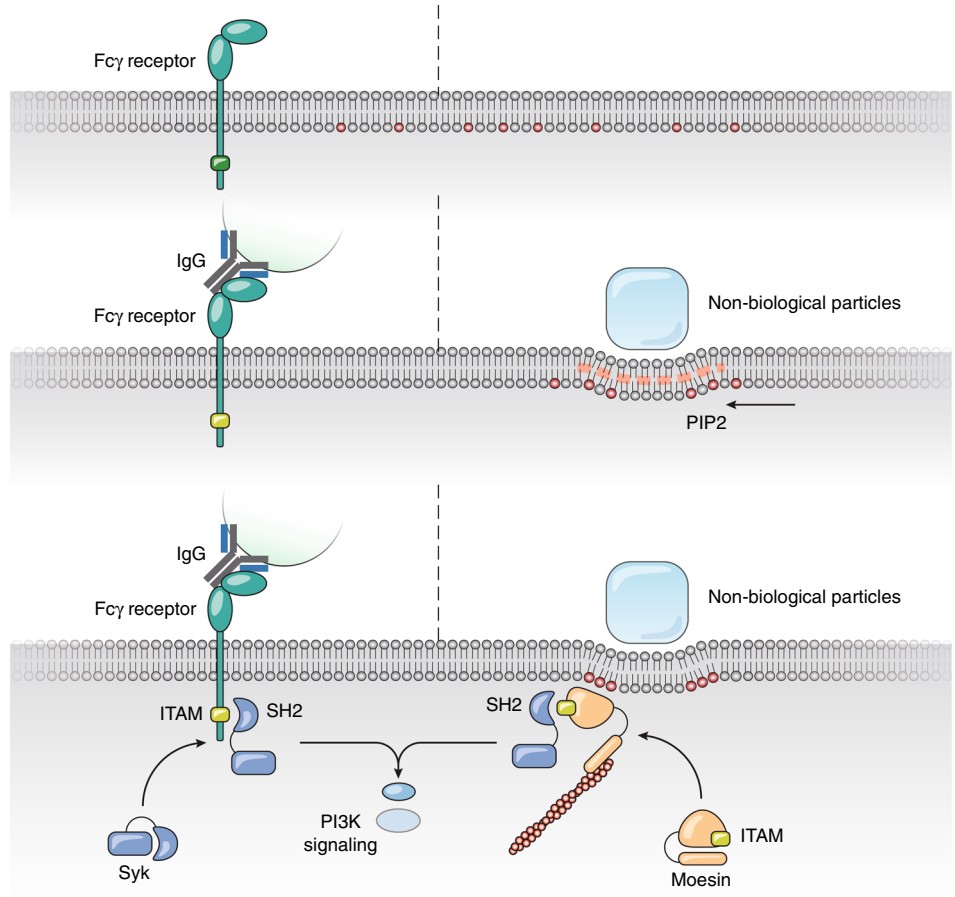

**Fig. 8** A summary model. A scheme of potential model how FcR and PIP2 redistribution-based mechanisms can share the entire signaling cascade downstream of ITAM for phagocytosis. When solid particles are engaged with plasma membrane, the binding induces membrane curvature change and sorting of PIP2 at the site of contact. This leads to the membrane recruitment of Moesin and its activation. Binding of Syk to Moesin consequently leads to all downstream phagocytic signaling and actin polymerization. On the other hand, during receptor-mediated phagocytosis, opsonized particles engage Fc receptors on the cell surface; ITAM of FcR becomes phosphorylated. This leads to the recruitment of Syk and all downstream phagocytic signaling identical to the lipid-based phagocytosis

AlexaFluor405-Streptavidin for 30 min on ice without permeabilization, thus differentiating uninternalized (with fluorescent label) beads from internalized (without fluorescent label) beads.

**Passive adsorption of proteins onto polystyrene beads**. Proteins were coated on polystyrene beads per Polysciences technical data sheet 238E. Briefly, 0.5 ml of the stock beads were washed with 0.1 M borate buffer (pH = 8.5) three times before incubation with 400 µg proteins overnight at room temperature with gentle end-to-end mixing. Beads were washed with PBS buffer three times after being centrifuged for 10 min. Finally, beads were resuspended in 1 ml PBS buffer and stored at 4 °C. For biotin-label, 0.5 ml of the stock beads were coated with 200 µg biotin-BSA; for IgG-label, beads were coated with 200 µg BSA and then with 200 µg mouse anti-BSA to expose the Fc fragment.

**Atomic force microscopy-based single cell force spectroscopy**. Experiments were performed as previously described using a JPK CellHesion unit[15]. In brief, a clean cantilever was placed on top of a glass block holder. Using a video camera attached to AFM, a laser was focused on the cantilever and properly aligned with respect to the position detector. Suitable silica crystals or polystyrene beads were placed directly onto the surface of a glass coverslip (diameter 2.4 cm); equal parts of the AB epoxy were mixed together and spread thinly at the side of the same glass coverslip. The clean cantilever gradually was moved to approach the glue so that only the very edge of the tip was over the glue. It was then lowered until a change in vertical deflection was observed and the tip had visible glue drawn up. Then the cantilever was pulled away from the glue until an observable snapping motion of the tip was observed, and was moved to individual silica crystals or polystyrene beads. The cantilever with glue was slowly lowered until the tip was in focus, and another deflection of the laser would be observed once the tip was in contact with the target. When raising the cantilever, the silica crystal or polystyrene bead on the glass coverslip should disappear if there was a successful attachment. To be certain

that the target was glued to the tip, it was also examined under a compound light microscope. Functionalized cantilever tips were left to harden in 37 °C for 30 min or room temperature overnight for further curing.

All instruments of AFM were turned on 30 min before single cell force spectroscopy (SCFS) experiments to ensure thermal equilibration and reduce drift during experiments. The coverslip seeded with DC2.4 cells was mounted onto a customized holder for AFM. The laser was focused on the back of the cantilever and aligned according to the guidelines of the AFM manufacturer. To calibrate the cantilever, a force–distance curve was recorded. Cantilever sensitivity and spring constant were determined using the routines incorporated in the AFM control software. For force reading, the cantilever was retracted approximately 50 µm from the surface. The apex of cantilever was manually positioned directly above a suitable cell. All measurements were performed with the JPK CellHesion 200 in relative force feedback contact mode (constant force 1.5 nN, contact time 5 s, pulling length 50 µm, and constant speed 5 µm s$^{-1}$). Force curves were normalized and analyzed using JPK data processing software. Maximum binding forces were calculated and plotted over time.

**Western blotting and co-immunoprecipitation assays**. For western blot analysis, in most experiments, $10^7$ cells were lysed with RIPA buffer. Total proteins were subjected to SDS-PAGE and immunoblotted with indicated antibodies. For co-immunoprecipitation, appropriate antibodies were incubated with protein A-Sepharose beads for 1 h at room temperature, followed by further incubation with $10^7$ cell lysates for 4–5 h at 4 °C. Immunocomplexes were washed four times with RIPA buffer and resolved by western blot analysis.

**Immunofluorescence**. Cells were fixed with 4% PFA for 10 min at room temperature followed by permeabilization with 0.1% Triton X-100 for 3 min at room temperature. After being washed three times, cells were incubated with primary antibodies at 4 °C overnight. Cells were washed again three times and incubated

with secondary antibodies or Alexa568-Phalloidin at 37 °C for 30 min. Cells were mounted with ProlongGold (Thermo P36975) mounting media and cured overnight before imaged by a microscope.

**Fluorescence microscopy.** Structured illumination microscopy (SIM) was performed on Carl Zeiss LSM780, an ELYRA super resolution microscope. Confocal microscopy was carried out on Nikon A1R+, a resonant scanning confocal system. Widefield microscopy was done on an Olympus IX-73 microscope or a home-built TIRF microscope.

SIM images were processed first with Zen software for calculation and channel alignment. Confocal images were processed and deconvolved with Nikon NIS Elements software. Images from an Olympus IX-73 microscope were deconvolved with CellSens software. Images from the home-built TIRFM were collected with MicroManager and processed in FIJI and deconvolved with Autoquant X. Line-profile analysis was done with Autoquant X. Kymograph analysis and all fluorescence intensities were measured with FIJI. 3D reconstruction of fluorescence images was done with Imaris 7.2.3.

**GPMV generation, modification, labeling, and observation.** GPMV were made according to Sezgin et al.[61]. with modifications. In short, cells were first cultured in a 35-mm culture dish overnight to reach 70% confluency. Before vesiculation, cell surface was biotinylated with an EZ-Link Sulfo-NHS-LC-Biotin kit (Thermo 21327). This step was added to provide extra adherence of GMPV to PDMS patterns coated with NeutrAvidin. Biotinylated cells were washed twice with 1 ml and then incubated with 1 ml freshly prepared 25 mM PFA/2 mM DTT at 37 °C for 60 min. GPMV were concentrated by leaving the suspension in a 1.5 ml Eppendort tube for 30 min. GPMV were labeled with Bodipy FL-PIP2, TopFluor TMR-PC, and TopFluor TMR-PE at 1:50 dilution on ice for 40 min. Labeled and biotinylated GPMV were then incubated with NeutrAvidin-coated PDMS patterns for 30 min at room temperature. Samples were observed with a Nikon A1R+ confocal microscope.

**Polystyrene microparticles with tunable rigidity.** Styrene (ST) and divinylbenzene (DVB, 55 wt%) were obtained from Dongda Chemical Engineering Group Co. Poly(vinyl alcohol) (PVA-2 17, degree of polymerization 1700, degree of hydrolysis 88%) was ordered from Kuraray (Japan). Benzoyl peroxide (BPO, with 25 wt% moisture content, reagent grade) was bought from Beijing Chemical Reagents Company. Fast Membrane Emulsifier (FM0210/500 M) and microporous membrane were provided by Senhui Microsphere Tech (Suzhou) Co., Ltd.

The premix membrane emulsification technique and the FMEM-500M equipment were used to prepare O/W (oil in water) emulsion. Different percentages of the crosslinking agent were used to tune the PST microparticle rigidity. In a typical process, 9 g ST as monomers, 0.43 g BPO as initiator (4.3 wt% of total monomers), and 1 g DVB or other dosages (10–90 wt% of total monomers) were commingled together and used as the oil phase. Deionized water (150 ml) containing 1.0 wt% PVA was used as the water phase. The oil phase and the water phase were mixed by low-speed stirring and poured into the premix silos. Then this coarse emulsion was extruded through the microporous membrane under a nitrogen pressure. The preliminarily emulsified emulsion was extruded through the same membrane as coarse emulsion in the next pass. The final emulsion was obtained after repeating the above process for three times. The obtained emulsion was transferred to a four-neck glass separator flask equipped with a semicircular anchor type blade, a condenser, and a nitrogen inlet nozzle. The emulsion was bubbled with nitrogen gas for 1 h. Then, the nozzle was lifted and the temperature was increased to 70 °C gradually. The polymerization was carried out for 20 h under nitrogen atmosphere. The prepared uniform microparticles were washed with hot water and then ethanol for four times each.

**Nanoindentation.** The rigidity of single polystyrene microparticles with different degrees of crosslinking was probed by an atomic force microscope (Veeco BioScope Catalyst, USA). Briefly, the microparticles or bead solution diluted with ethanol was dipped on the surface of a clean glass substrate. The beads adhered to glass surface primarily due to the capillary force. The topography of the surface was first scanned in tapping mode with a stiff probe (Veeco TESPA, spring constant 20–80 N m$^{-1}$, resonant frequency 230–320 kHz) and the location of individual beads was then identified for force-indentation displacement probing. The force applied for indentation ranged from 100 to 1500 nN. The observed indentation depth was less than 10 nm, about 0.3% of the size of beads. The same beads were probed three times and at least 12 beads were investigated for each sample. The rigidity (Young's modulus) of these beads was obtained through indentation analysis based on the Herzt model with a Poisson ratio of 0.33 for bulk polystyrene using the native Veeco analyzing software.

**PDMS patterns.** Micropatterned array was fabricated with conventional semiconductor microfabrication and soft lithography. Briefly, a fused silica photomask carrying the designed micropatterns was first made through a photolithography process (Beijing Zhongke Shengze Technology Development Co., Ltd.). The Si

micropatterned mold was then fabricated by high-resolution photolithography and dry etching techniques (Department of Microelectronics, Peking University).

PDMS micropatterned array was generated by replica molding. Briefly, PDMS and the curing agent (SYLGARD® 184) were combined in a 3:1 ratio and mixed thoroughly using a pipette for 15 min. PDMS mixture was degassed for 1 h under vacuum. A thin layer of degassed PDMS mixture was cast onto the chlorotrimethylsilane (MACKLIM)-coated resist master by spinning at 1000 rpm for 1 min. After overnight incubation at 60 °C, individual hardened patterns (1 mm × 1 mm) were peeled off and stored at room temperature. To prepare for GPMV/pattern contact experiment, patterns were pre-coated with NeutrAvidin (A2666, Invitrogen) and stored at −20 °C.

**Phylogenetics.** The amino acid sequences (RefSeq) of each protein in *Homo sapiens* (NP_001129691.1, NP_004097.1, NP_002435.1, NP_003168.2, NP_002638.2, NP_932170.1, NP_003323.1) were used as probes to find their orthologs from the model organism database by NCBI protein BLAST (https://blast.ncbi.nlm.nih.gov/). Leaving all of the other settings at their default values, the top RefSeq hits of each species were selected to perform multiple sequence alignment and construct phylogenetic trees. The maximum likelihood phylogenetic reconstruction of the amino acid matrices were performed using the PhyML 3.0 web server (http://www.atgc-montpellier.fr/phyml/)[62]. The amino acid substitution models were selected using the Akaike Information Criterion by the SMS[63] integrated in PhyML 3.0 web server. Each tree searching was performed using ten times of random starting tree and SPR algorithm. The branch supports were calculated using one hundred bootstrap samples.

To present the relative evolutionary rates of proteins among different groups of organisms, a time axis was shown along with the phylogenetic tree in the unit of million years before present, with *y*-axis showed number of substitutions per site compared to the amino acid sequence of the homologs in *H. sapiens*, so the slopes of branches suggest amino acid substitution rates. The species divergence times were used to represent the divergence time between orthologous amino acid sequences, which were from the estimated time derived from previous studies in the TIMETREE website (http://www.timetree.org/)[64–67]. In the phylogeny of paralogous ZAP70 and Syk, the tree root age was calculated by amino acid substitution rates on both ZAP70 and Syk clades respectively based on the molecular clock.

**Statistical analysis.** GraphPad Prism 7 (Graphpad Software) was used to make graphs and to perform statistical analysis. An unpaired, two-tailed Student's *t*-test was conducted to compare two samples. When comparing multiple samples, one-way analysis of variance (ANOVA) models was used followed by post hoc Scheffé test with STATA 14 (StataCorp LLC). These tests were considered not significant when $p \geq 0.05$ (ns), significant when $0.01 \leq p < 0.05$ (*), very significant when $0.001 \leq p < 0.01$ (**), and extremely significant when $0.0001 \leq p < 0.001$ (***) or when $p < 0.0001$ (****). For correlation analysis, Spearman's rank correlation coefficient *r* and *p*-value were calculated with GraphPad.

## Data availability
The data that support the findings of this study are available from the corresponding author on request.

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

## Acknowledgements

We thank Dr. Dawn ME Bowdish for providing SR-A6$^{-/-}$ SR-A1$^{-/-}$ DKO bone marrows. We thank Dr. Shu-Jin Luo of Peking University for her help with the

phylogenic analyses. We thank Dr. Li Yu of Tsinghua University for critical review of this manuscript. We thank Microscopy and Imaging Facility and Live Cell Imaging Facility of the University of Calgary for technical assistance. Y.S. is supported by the joint Peking-Tsinghua Center for Life Sciences, the National Natural Science Foundation of China General Program (31370878), State Key Program (31630023) and Innovative Research Group Program (81621002), and by grants from the US NIH (R01AI098995), and the Canadian Institutes for Health Research (MOP-119295).

## Author Contributions

Libinb M. and Z.T. performed all the experiments unless noted otherwise. Lin M. advised on phytogenic analysis. Y.H. performed siRNA screening of knockdown cells. N.K. provided technical assistance on AFM-related experiments. J.C. performed AFM-SCFS. H.R. assisted in imaging analysis. W.W. and F.G. provided technical assistance on polystyrene microparticles synthesis with guidance from G.M. B.W. coordinated the manufacture of silicon wafer with guidance from Y.D. Y.S., Libing M. and Z.T. designed all experiments and analyzed data. Lin M., Z.T. and T.X. contributed to manuscript preparation. T.X. performed nanoindentation work and supervised the project. Y.S. wrote the manuscript with inputs from T.X and M.W.A.

## Additional information

**Competing interests:** The authors declare no competing interests.

