## [Peer Review File · Nature Communications]

Reviewers' comments:

Reviewer #1 (Remarks to the Author):

Mu et al

This very interesting manuscript by Mu et al sets out to identify cryptic ITAM motifs in proteins that might recruit Src family members during receptor-independent phagocytosis. The authors used bioinformatics to identify moesin as a candidate and show that moesin knockdown impairs phagocytic particle uptake in macrophage cell lines. This activity seems to be rescued by the moesin FERM domain alone, which can also co-IP Syk, the non-receptor tyrosine kinase that associates with ITAM motifs in the Fc receptor. The authors then argue that PIP2 clustering can be induced by mechanical deformation/solid particle contact applied to the membrane alone, which in turn recruits and activates moesin for Syk binding and phagocytic signaling/initiation. The authors argue that this activity is the evolutionarily ancient 'innate' mechanism of receptor-independent phagocytosis.

This is a very interesting paper that not only suggests a novel mechanism of ERM activation (mechanical membrane deformation – induced PIP2 clustering) that might be applicable to many other situations, but a novel receptor-independent mechanism of phagocytosis. My major concerns, as detailed below, are the need for improved experimental support for the conclusion that the actin-linking function of ERMs is not necessary for this activity, a more careful evaluation of ezrin and radixin in these studies, as it pertains to redundancy/compensation or specificity – a key issue for the field, and an integration of this new data with the authors' own existing publications. With these additions/modifications, this would make a nice publication in Nature Communications.

Specific points:

1) The identification of an ITAM motif in ERM proteins and Syk-moesin interaction have already been described (Urzainqui et al) and this work is only cited peripherally at the end of the manuscript. Urzainqui et al did not examine phagocytosis and focused on the PSGL-1 receptor, but their paper should be cited in the results when moesin is identified as an ITAM containing candidate and Syk-interactor.

2) Beginning in Fig 1b – presumably 'NS' in the x-axis legend is a control but this is never defined.

3) Fig. 1D, S1 it is highly unlikely that the anti-ERM antibodies used cleanly distinguish between the three (none of the epitopes have been defined and they have not been rigorously tested for this). The field suffers from their misuse and it is a very open question as to whether the ERMs are truly functionally redundant. Moesin does look to be the most enriched, consistent with mRNA expression data. However, the authors should at the very least examine shEzrin, shRadixin and shMoesin immunoblots and cells (by IF) for the other family members (shRadixin is not a compelling knockdown anyway) – so that the data they present is not misleading or taken by others as accurate use of these antibodies. Fig 1d is misleading as it seems to suggest that the activity is specific for moesin – this should be avoided. The total ERM burden will be important in the rescue experiments as ERMs have

been argued to oligomerize (see below as well).

4) The linker and C-terminal fragments of ERM proteins almost certainly do not localize to the plasma membrane in Cos-1 cells; therefore, it cannot be argued that they do not associate with Syk on the basis of a failure to co-IP. The statement: 'This result indicated that FERM domain-mediated phagocytosis was a consequence of its association with Syk' is not shown by this data; it is correlative.

5) The question of whether the actin-binding activity of moesin/ERM is important for this is critical. The authors claim that the FERM domain-only rescue indicates that actin binding is not required. However, moesin is not completely knocked down in these experiments and it is not known whether ezrin and radixin is upregulated to compensate; this is critical given the published propensity of ERMs to heterodimerize or oligomerize and the published propensity for them to be compensatorily upregulated in mouse knockouts. Does the FERM domain only rescue cytoD-treated cells? The cytoD experiment is shown in Fig 1e but never mentioned in the text.

6) The authors should discuss the ITAM motif localization in terms of the well-established crystal structure of moesin and the association of the FERM domain with the C-terminal tail. Is it masked in the closed conformation and where is it relative to the established PIP2-binding domain?

7) The evolutionary analysis is interesting but cannot be used in place of data in support of the authors' model. This section should be shortened at least. A phylogenetic analysis of ERM proteins has been published and should be cited. The rescue with worm and fly orthologues is a nice point in the paper.

8) In the discussion the authors refer to their own previous conclusion that lipid rafts are the initiators of receptor-independent phagocytosis and now argue that mechanical/solid structure induced lipid raft aggregation induces PIP2 aggregation. Testing this would reconcile their own differing models – are lipid rafts co aggregated by the methods shown in figure 2? Does depletion of membrane cholesterol impair moesin-mediated phagocytosis?

9) It is gently but strongly suggested that the authors carefully scour the manuscript for grammatical and spelling errors. The meaning of sentences is often lost for incorrect phrasing.

10) The authors should check for missing references – several background statements are without.

Reviewer #2 (Remarks to the Author):

Mu et al. demonstrates that an ITAM harboring protein, Moesin, is involved in phagocytosis in the absence of FcR signaling. The authors found that PIP2 is autonomously accumulated at site of solid structure binding, where recruiting Moesin and Syk to activate signaling. Further phylogenic analysis identified that PI3K and Syk appeared about 0.8 billion years ago, which is much earlier than the dawn of FcR receptors, implying that Moesin-based phagocytosis is more conserved form of solid particle uptake pathways.

Overall, this manuscript is novel and interesting, though following points need to be addressed.

Specific comments

1. The idea that a binding force exerted on a membrane from a solid particle inducing phagocytosis is interesting. While the authors have speculated that lipid raft aggregation is responsible for initiating receptor-independent phagocytosis, the article presented here would have been more convincing if this was experimentally proven. Specifically that the lipid rafts aggregation causes PIP2 aggregation.
2. The idea that just by having an ITAM motif, a surface molecule is converted into a primitive phagocytic receptor need to be further proven and developed with more examples. Again, what signals, specific to ITAM are provided or received, that, in the absence of Moesin can induce phagocytosis? It would be interesting to uncover the protein responsible for this.
3. The authors have not shown that Moesin directly binds to accumulated PIP2. This might be the weakest link in the paper. Co-localization data does not prove direct binding.
4. Phylogenetic trees shown in Fig. 4 are intriguing, though analysis of additional ITAM-containing proteins is needed to strengthen the authors assumption.
5. Authors should clearly state the purpose of using MSU in the experiment in the main text.
6. Fig3a, b suggests that a functional ITAM can trigger phagocytosis, independent of moesin. However, the phosphorylation of the ITAM is still required for this particular moesin-independent phagocytic process. What drives the phosphorylation? It is known that Lck is responsible for the phosphorylation of ITAM in CD3 and ζ chains in the TCR. Is it possible that Lck phosphorylation might be occurring in this scenario? Is this phenomena only specific to the CD4 chimera or can other surface molecules be converted into ITAM chimeras and this sequence of events repeated?
7. The authors switch between DC2.4 (fig S3d) to RAW264.7 cells (fig S3e). Is there a reason for this?
8. Fig S3h. The authors switch to murine BMDM. Is this change specific with regard to SR-A6/SR-A1 experiments? If so, why?

Minor comments

1. Fig 1c. Seven clones were shown in the immunoblot. However, the phagocytosis assay shows only two. Which clones were used?
2. Lines 72-75, citations are required.
3. Fig 2d, S2b. PH-mCherry not mentioned in the Materials and Methods section. Authors should explain where this plasmid was obtained from, indicating the catalogue number. If

the authors generated the plasmid, it should also be included in the Materials and Methods section. It should be noted that in general, the PH-domain can bind both PIP2 and PIP3. Therefore, it should be specified how specific the PH-domain is to the former and latter.

Reviewer #3 (Remarks to the Author):

Mu et al describe the finding that the actin binding protein, moesin, functions to activate the Syk tyrosine kinase during uptake of solid particles by phagocytic cells. Specifically, interaction of particles with the external plasma membrane (either directly or through receptor-mediated binding) leads to accumulation of PIP2 in the inner portion of the plasma membrane, resulting in activation of moesin, which through an ITAM-like sequence associates with Syk kinase to mediate downstream signaling leading to actin polymerization and uptake of the particle. Knockdown of moesin reduces solid particle (latex bead) uptake; re-expression of the moesin FERM domain (which has the ITAM like sequence) rescues particle uptake, while expression of an ITAM-point mutant of the moesin FERM domain is inactive. PIP2, moesin and Syk kinase all co-localize around the particle during uptake. Moesin-mediated signaling of may also be involved in scavenger receptor mediated particle uptake.

This is an interesting and well done paper. The role of ERM proteins linking to actin to various receptors is well described, but the idea that moesin is signaling intermediate is novel and interesting. There are several questions that remain to be addressed.

1) Moesin knockdown in DC2.4 cells reduces latex bead uptake from ~80% to ~30%. Cyto D treatment goes to ~5%. What accounts for the remaining bead uptake? What is the efficiency of reduction of bead uptake with Syk knockdown in the same cells? Would this experiment be done better with CRISPR deletion of moesin?

2) I was very surprised that transfection of the FERM domain of moesin alone could rescue. How is that possible, since there would be no way to link polymerization of the actin cytoskeleton to the plasma membrane. Do the authors have some idea why this works? Some discussion of this is needed.

3) Most importantly, there is no data or discussion of phosphorylation of moesin. There is a phosphorylation event shown in the summary model, but it is not demonstrated in the data. Does latex bead binding to the membrane induce phosphorylation of moesin? Is this mediated by Src-family kinases? How would they sense association of a latex bead to the plasma membrane? Some experiments examining the kinetics of moesin phosphorylation are needed.

4) Why are Cos-1 and Cos-7 cells phagocytic but HEK293T and NIH3T3 not? Is it expression of Syk?

5) Most importantly, I believe that Grinstein's lab demonstrated many years ago that during

FcR-mediated phagocytosis that PIP2 is EXCLUDED from the base of the phagocytic cup to allow actin polymerization of the arms of membrane around the IgG opsonized particle. The authors should discuss why PIP2 accumulation is required for inert particle uptake and seems to be excluded for FcR mediated phagocytosis.

6) I believe that MSU uptake by macrophages is very pro-inflammatory while latex bead uptake is not. Is there a difference in the proximal signaling responses (PIP2 accumulation, Syk activation, moesin phosphorylation) that may help explain this?

Otherwise this is an interesting report that helps explain the mechanisms of inert particle engulfment by phagocytic cells.

We would like to thank all three reviewers for their high remarks and their enthusiasm. As this work is a unique combination of immune signaling, membrane biology and evolution, we were a little concerned whether our goal to unify these analyses to trace back to the early point of immune signaling was risky. To our profound delight, all reviewers were very receptive and encouraging. With that said, the issues raised are still critical to improve the quality of this manuscript. Except for one place where we asked for the clarification of the question, we tried to address all the issues with additional experiments, which are presented either in the manuscript itself or in the point by point document for reasons outlined at specific places. All major changes are marked in purple.

Reviewer #1 (Remarks to the Author):

This very interesting manuscript by Mu et al sets out to identify cryptic ITAM motifs in proteins that might recruit Src family members during receptor-independent phagocytosis. The authors used bioinformatics to identify Moesin as a candidate and show that Moesin knockdown impairs phagocytic particle uptake in macrophage cell lines. This activity seems to be rescued by the Moesin FERM domain alone, which can also co-IP Syk, the non-receptor tyrosine kinase that associates with ITAM motifs in the Fc receptor. The authors then argue that PIP2 clustering can be induced by mechanical deformation/solid particle contact applied to the membrane alone, which in turn recruits and activates Moesin for Syk binding and phagocytic signaling/initiation. The authors argue that this activity is the evolutionarily ancient ‘innate’ mechanism of receptor-independent phagocytosis.

This is a very interesting paper that not only suggests a novel mechanism of ERM activation (mechanical membrane deformation – induced PIP2 clustering) that might be applicable to many other situations, but a novel receptor-independent mechanism of phagocytosis. My major concerns, as detailed below, are the need for improved experimental support for the conclusion that the actin-linking function of ERMs is not necessary for this activity, a more careful evaluation of ezrin and radixin in these studies, as it pertains to redundancy/compensation or specificity – a key issue for the field, and an integration of this new data with the authors’ own existing publications. With these additions/modifications, this would make a nice publication in Nature Communications.

We thank the reviewer for the accurate summarization of the work and his/her encouragement

Specific points:

1) The identification of an ITAM motif in ERM proteins and Syk-Moesin interaction have already been described (Urzainqui et al) and this work is only cited peripherally at the end of the manuscript. Urzainqui et al did not examine phagocytosis and focused on the PSGL-1 receptor, but their paper should be cited in the results when Moesin is identified as an ITAM containing candidate and Syk-interactor.

This is true, and the due credit has been given at the point where Moesin was found to be the dominant signaling transducer from the membrane (Urzainqui et al., 2002)

2) Beginning in Fig 1b – presumably ‘NS’ in the x-axis legend is a control but this is never defined.

This is our omission. In this version, all “NS” are defined in the figure legend.

3) Fig. 1D, S1 it is highly unlikely that the anti-ERM antibodies used cleanly distinguish between the three (none of the epitopes have been defined and they have not been rigorously tested for this). The field suffers from their misuse and it is a very open question as to whether the ERMs are truly functionally redundant. Moesin does look to be the most enriched, consistent with mRNA expression

data. However, the authors should at the very least examine shEzrin, shRadixin and shMoesin immunoblots and cells (by IF) for the other family members (shRadixin is not a compelling knockdown anyway) – so that the data they present is not misleading or taken by others as accurate use of these antibodies. Fig 1d is misleading as it seems to suggest that the activity is specific for Moesin – this should be avoided. The total ERM burden will be important in the rescue experiments as ERMs have been argued to oligomerize (see below as well).

We provide evidence of additional IF images to show the Moesin accumulation to the beads as detected by our anti-Moesin antibody under Ezrin and Radixin knockout (Fig. S1i). There, the accumulation of signals detected by anti-Moesin antibody is not significantly altered by Ezrin or Radixin knockdown. The WB shows the antibodies of ERM are specific enough to detect each protein (Fig. S1j). There was little compensation when we knocked down these proteins. Regarding Fig 1d, despite the new data and evidence provided by the phagocytosis assay, we softened our tone in stating the connection between Moesin and membrane binding-mediated phagocytosis.

4) The linker and C-terminal fragments of ERM proteins almost certainly do not localize to the plasma membrane in Cos-1 cells; therefore, it cannot be argued that they do not associate with Syk on the basis of a failure to co-IP. The statement: ‘This result indicated that FERM domain-mediated phagocytosis was a consequence of its association with Syk’ is not shown by this data; it is correlative.

We had a little difficulty understanding this question. For the molecules involved here, Syk is a cytosolic protein that only gets recruited to the plasma membrane once ITAMs are phosphorylated. FERM domain can be recruited if there is an appropriate protein target (CD44, PSGL etc) or accumulated PIP2 (step 1). Once FERM domain moves to the membrane, its ITAM is phosphorylated by Src family members (step 2). Syk then docks on to it and mediates downstream phagocytic event (step 3). Because our data show that only the FERM domain showed IP with Syk, our statement was to indicate the critical event of FERM domain/Syk binding. We guess that the reviewer’s comment may indicate 1. Our statement is too strong to link FERM/Syk binding event to phagocytosis. 2. There is a chance that Syk can be associated with the linker and the C domain if we first moved Syk to the membrane. As we failed to do so, the absence of such a binding was expected. 3. Because Moesin is a whole protein, the other two domains can be physically proximal to Syk as a consequence of FERM domain binding. 4. Our statement simply ignore the potential contributions to phagocytosis by the linker and C domain as Moesin may use FERM domain as the signal transducer and the C domain for cytoskeletal attachment (and the latter is also critical), a message reiterated in Q5. 5. If we design an experiment to attach the linker and the C-domain to the membrane, phagocytosis may take place with or without FERM domain. Certainly, there are other possibilities.

As a starter, we revised the statement, merely suggesting the Syk can bind to FERM domain, its biological implications are left to the later discussion (Fig. S1m, n, and p). In the previous version, we also discussed the scenarios where the linker and C domains intrinsic to the FERM domain may or may not be important for phagocytosis (Fig. 1e). The discussion is expanded in this version.

These guesses likely miss the intended message of the reviewer. Rather than trying more experiments with this murky understanding, we wonder if the reviewer can rephrase the question so that we can address the issue head on.

5) The question of whether the actin-binding activity of Moesin/ERM is important for this is critical. The authors claim that the FERM domain-only rescue indicates that actin binding is not required. However, Moesin is not completely knocked down in these experiments and it is not known whether

ezrin and radixin is upregulated to compensate; this is critical given the published propensity of ERMs to heterodimerize or oligomerize and the published propensity for them to be compensatorily upregulated in mouse knockouts. Does the FERM domain only rescue cytoD-treated cells? The cytoD experiment is shown in Fig 1e but never mentioned in the text.

We would like to break down this question into smaller ones. First, regarding the experiment under Cyto-D treatment, we take this as a friendly jab from the reviewer to reveal the flaw in our logic – it would be absurd if we can manage to observe phagocytosis under actin polymerization blockage. In this experiment, the Cyto-D was used as negative control only and this has been indicated in the figure legend (Fig. 1e).

The real question is therefore whether in addition to FERM/Syk binding, the intrinsic actin binding activities are still critical, or can the latter be provided in trans. In our data, under Moesin KD, mere provision of FERM domain can rescue, it would therefore suggest that adding an ITAM containing sequence under the inner leaflet (FERM domain will bring this sequence to the membrane in the absence of C-ERMAD fold back autoinhibition) is sufficient to activate phagocytosis. This construct does not have the actin binding capacity. Under Moesin KD, we provide new data here that Ezrin and Radixin did not go up (Fig. S1i and j), however, this does not exclude their potential role as the linker critical for phagocytic cup formation. Ideally, to definitively rule out the C-ERMAD involvement, we need to produce a complete Moesin deficient cell line than providing the FERM domain construct. Unfortunately, when we performed such a KD with increased dose, the treated cells no longer attached to the plate (insert 1), rendering phagocytosis assay difficult to read. Even if such experiment was successful, it would still argue that the linker function can be provided in trans. We made this notion clear with a relatively detailed discussion in the text. This issue is also linked to the presence of other non-Moesin mediated, receptor independent phagocytosis, which will be discussed later.

Insert 1. Complete knockout of Moesin impaired cell adhesion.

Left: Total cell lysates of Moesin KD clone, NS control cells, Moesin KO clone with Crispr and wildtype DC2.4 cells were subjected to immunoblotting with the indicated antibodies. Right: Bright field image of indicated cells under cell culture condition, showing Moesin KO cells had weaker surface adhesion.

6) The authors should discuss the ITAM motif localization in terms of the well-established crystal structure of Moesin and the association of the FERM domain with the C-terminal tail. Is it masked in the closed conformation and where is it relative to the established PIP2-binding domain?

We used data from a paper and discussed the established understanding regarding the relative position of each domain for activated and autoinhibited Moesin (Edwards and Keep, 2001; Hamada et al., 2000; Li et al., 2007; Pearson et al., 2000). We also added a discussion. Because the discussion is long in its entirety, an abbreviated version is provided in the manuscript. The full discussion below is to show the reviewer that structural insights of Moesin conformational change have been reported in the literature.

This question is deconstructed into 3 sub-questions.

1. Where is ITAM in Moesin?

According to the primary structure (sequence), ITAM, specifically Y191 and Y205, is localized within the FERM domain of Moesin between F2 (residues 96- 195) and F3 (residues 204-296) subdomains.

2. Is ITAM of Moesin masked in the closed conformation?

Yes, very likely. Crystal structure analysis of full-length Moesin isolated from cells revealed that the FERM domain is masked by C-ERMAD domain in the closed conformation (Edwards and Keep, 2001; Pearson et al., 2000). Specifically, crystal structure of Moesin in its closed conformation from *Spodoptera frugiperda* (PDB: 211K) showed that Y191 and Y205 on Moesin (red) on the FERM domain (cyan) is masked by C-ERMAD (yellow) (insert 2). Note: the figures below were generated by us with critical residues indicated from the original figure produced from a previous paper (Li et al., 2007).

Insert 2. The Y191 and Y205 of Moesin ITAM are masked in the closed conformation.

Crystal structure of the *Spodoptera frugiperda* Moesin (PDB: 211K) in a closed state. F1 F2 and F3 shows the three lobes of FERM domain. The Coiled coil α -helix and linker region are represented in gray, and the yellow part shows C-terminal ERMAD. The two red residues are the key tyrosines of ITAM motif, Y191 and Y205, which are masked by the C-ERMAD.

3. Where is the established PIP2-binding domain on Moesin?

Binding of Moesin to PIP2 is mediated by positively-charged residues on the FERM domain. First, K63 and K278, together termed the “POCKET” (red), are shown to bind PIP2. They are in a cleft between F1 and F3 subdomains (Hamada et al., 2000) (insert 3). Second, K253/K254 and K262/263, together termed the “PATCH” (purple). They reside in F3 subdomain. Accessibility to PIP2 in autoinhibited Moesin differs between the POCKET and PATCH. The PATCH on autoinhibited Moesin is more accessible to PIP2 because of its availability and its favorable surface potential. By contrast, the POCKET in autoinhibited Moesin is covered by the autoinhibitory linker FLAP. Since FLAP is a steric hindrance for PIP2 access to the POCKET and its strong negative charges reverses the positive electrostatic potential around the POCKET, POCKET is much less accessible by PIP2 in autoinhibited Moesin.

It was proposed that PIP2 binds and activates Moesin progressively. First, PIP2 transiently binds to the more accessible PATCH and initiates the release to FLAP. Next, the same PIP2 molecule stably binds to the now more accessible POCKET to complete conformational activation (Ben-Aissa et al., 2012; Hamada et al., 2000). Presumably, this can lead to the exposure of ITAM for activation and binding.

Insert 3. The critical residues of PIP2 binding on Moesin.

Crystal structure of the *Spodoptera frugiperda* Moesin (PDB: 211K) in a closed state. F1 F2 and F3 shows the three lobes of FERM domain. The coiled coil α -helix and linker region are represented in gray, and the yellow part shows C-terminal ERMAD. The two red residues, K63 and K278, termed the PIP2-binding ‘POCKET’; and the four purple ones, K253/K254 and K262/263, together termed the ‘PATCH’.

7) The evolutionary analysis is interesting but cannot be used in place of data in support of the authors’ model. This section should be shortened at least. A phylogenetic analysis of ERM proteins has been published and should be cited. The rescue with worm and fly orthologues is a nice point in the paper.

We cited two papers that discuss ERM evolution and the potential adaptation of ERM proteins towards the final unification with modern day phagocytosis (Bretscher et al., 2002; Golovkina et al., 2005). The phylogenetic analyses were performed by SJ Luo's lab at Peking University following standard protocols of comparative genetics labs. Although they cannot be used as biochemical evidence obtained from present day animals, this was a significant undertaking and should be able to give sound genetic connections among the tested species. As this report aimed at revealing these connections, we hope these results can also be accepted as data here. Regarding the placement of the phylogenetic data, as another reviewer intends to see a more complete analysis, we are not sure how to proceed. Here we performed a more detailed analysis and the data is presented in new Fig 4a, shortened version is also provided as a supplemental figure (Fig. S4d). They can be easily swapped. Can we ask the editor to decide their final placements?

8) In the discussion the authors refer to their own previous conclusion that lipid rafts are the initiators of receptor-independent phagocytosis and now argue that mechanical/solid structure induced lipid raft aggregation induces PIP2 aggregation. Testing this would reconcile their own differing models – are lipid rafts co aggregated by the methods shown in figure 2? Does depletion of membrane cholesterol impair Moesin-mediated phagocytosis?

The reviewer points a critical yet cryptic issue here – how do we link this finding to our previous model that lipid raft sorting initiates phagocytosis? In fact, establishing this link is the most important pursuit of our lab for the next few years. The mechanical forces at work are very complex and the lipid dynamics can only be resolved with superresolution imaging techniques such as STED/PALM which is very demanding work. We have produced some semi-mature data in that direction which forms the basis for several ongoing projects in the lab. Here we show as an insert using PDMS micropatterns to drive the colocalization of PIP2 into highly ordered membranes indicated by a lipid packing sensitive dye c-Laurdan. We observed preferential sorting of PIP2 into highly ordered membrane on RAW264.7 cells engaged with PDMS micropatterns (Fig. S2h). This figure suggests that the two events are related. Moreover, when we disrupt lipid raft via cholesterol extraction by methyl- β -cyclodextrin (M β CD), we found phagocytosis of polystyrene beads was impaired in a dose-dependent manner (insert 4). However, we would like to ask the reviewer's understanding of our desire not to show this preliminary data in this manuscript. We do not believe our data are complete enough to make any definitive statement at this moment.

Insert 4. The treatment of methyl- β -cyclodextrin (M β CD) impairs phagocytosis.

The phagocytosis assays were performed on DC2.4 cells with or without treatment by indicated concentrations of M β CD.

9) It is gently but strongly suggested that the authors carefully scour the manuscript for grammatical and spelling errors. The meaning of sentences is often lost for incorrect phrasing.

This manuscript has now been read by several colleagues and some grammatical and syntax issues have been corrected to the best of my capacity.

10) The authors should check for missing references – several background statements are without. Several additional references have been added to the introduction and discussion, they are marked in purple.

Reviewer #2 (Remarks to the Author):

Mu et al. demonstrates that an ITAM harboring protein, Moesin, is involved in phagocytosis in the absence of FcR signaling. The authors found that PIP2 is autonomously accumulated at site of solid structure binding, where recruiting Moesin and Syk to activate signaling. Further phylogenetic analysis identified that PI3K and Syk appeared about 0.8 billion years ago, which is much earlier than the dawn of FcR receptors, implying that Moesin-based phagocytosis is more conserved form of solid particle uptake pathways.

Overall, this manuscript is novel and interesting, though following points need to be addressed.

We thank the reviewer for the positive comment

Specific comments

1. The idea that a binding force exerted on a membrane from a solid particle inducing phagocytosis is interesting. While the authors have speculated that lipid raft aggregation is responsible for initiating receptor-independent phagocytosis, the article presented here would have been more convincing if this was experimentally proven. Specifically that the lipid rafts aggregation causes PIP2 aggregation.

Similar to the response to the comment by reviewer 1, we have initiated significant efforts to sort out the link between lipid raft motion in response to binding induced curvature change and PIP2 sorting. The aggregation of PIP2 towards lipid rafts has been suggested in a few previous papers (Johnson and Rodgers, 2008; Kim et al., 2007; Machnicka et al., 2014; Ponuwei, 2016), although the underlying mechanism is basically unknown. The critical issue here is how to couple the dynamic lipid domain mixing and demixing (time scale of mili second to seconds) with the slightly different dynamics of PIP2 accumulation (a result of both lipid motion and enzymatic metabolism). An equally important question is what type of curvature that ordered domains exert to favor PIP2 accumulation. We have established several advanced superresolution imaging and lipid half-life analyses, but the topic is far deeper and intermingled than we had imagined, so much so that the work needs to be divided into several graduate student projects. Thus far we can only provide the following information with certain degree of confidence 1. PIP2 preferentially sorts into highly ordered membranes, which is indicative of the presence of lipid rafts (Fig. S2h). 2. Removal of cholesterol abolishes PIP2 aggregation at the site of contact with beads (insert 5). 3. In the FERM domain mediated phagocytosis, the cholesterol disruption blocked bead uptake (insert 4). As the data are far from being systematic, we also request this reviewer's understanding of our hesitation to show all the data to the public at this moment.

Insert 5. PIP2 failed to sort into phagocytic cups with disruption of lipid rafts by MβCD

RAW 264.7 cells expressed with PIP2 sensor PH-PLCδ-GFP were treated with cholesterol extraction agent MβCD (10 mM, upper) or control (DMSO, lower) for 30 mins at 37 °C before further incubation with naked 3 μm polystyrene beads for 5 mins. Cells were fixed and PIP2 distribution was analyzed by confocal microscopy. The position of individual beads in contact with cell membrane is indicated with “*”. Scale bars, 5 μm.

2. The idea that just by having an ITAM motif, a surface molecule is converted into a primitive phagocytic receptor need to be further proven and developed with more examples. Again, what signals, specific to ITAM are provided or received, that, in the absence of Moesin can induce phagocytosis? It would be interesting to uncover the protein responsible for this.

First, we advocate a unified phagocytosis pathway proposed by us termed “SEP” or signaling equivalent platform (Shi, 2012) that suggests that the basic requirements for phagocytosis discussed in this report are at work in the absence of phagocytic receptors. The membrane based phagocytosis existed first with a complete signaling system. Other receptors that came later, such as FcRs, inherited the basic building blocks of this system but evolved to use more efficient receptors for the same purpose. Therefore, the main signaling components are the same. In this version we have figures to show that they are sensitive to the same set of inhibitors (Fig. S3b). From this analysis, we believe that the established FcR based signaling has already enlisted all the critical components for our pathway. Identifying ones that are unique to our system will be difficult.

We have also created two additional versions with ICAM-1 and CD8 (alpha) as the external binding sequences, each with or without a cytosolic Moesin ITAM. Ligation of each molecule was able to induce significant level of phagocytosis in Cos-1 cells (Fig. S3c and d). Of note, in 2006, Sergio Grinstein used CD44 as the external binding target to induce phagocytosis (Vachon et al., 2006). Although the premise was different, the implication remains that ligation of cell surface proteins can trigger phagocytosis (with undisturbed ERM). Importantly, similar to the finding of CD4 counterpart, the intracellular ITAM was required for phagocytosis under Moesin KD (Fig. S3c and d). Therefore, external ligation coupled to cytosolic Moesin can trigger phagocytosis. On the other hand, if this ITAM is directly linked to a “receptor”, the chimeric construct can signal for phagocytosis independent of intracellular Moesin.

3. The authors have not shown that Moesin directly binds to accumulated PIP2. This might be the weakest link in the paper. Co-localization data does not prove direct binding.

The binding between PIP2 and Moesin has been tested with several different setups by other labs (Hao et al., 2009; Hirao et al., 1996; Lubart et al., 2018; Yonemura et al., 2002). We therefore did not design a new experiment to prove the connection again in the last version. Per this reviewer's request, we set up a FRET based system using Moesin as the donor and the PIP2 containing vesicle as the recipient (Ben-Aissa et al., 2012). Although we have confirmed the finding (insert 6), this is not new data and our design is copied from that paper. Please allow us to only show the data to the reviewers but not in the manuscript lest leaving the false impression that we developed the methodology.

Insert 6. Interaction between PIP2 and Moesin

(left) After correction for direct acceptor excitation, the acceptor emission was observed due to FRET upon donor excitation (PIP2 line at 580 – 610 nm), indicating that Moesin indeed binds to PIP2 as expected. This is however not observed with GFP only samples (right).

Method for Moesin to membrane PIP2 FRET assay

The protein to membrane FRET assay was designed based on the well-established approach (Lai et al., 2010). Briefly, large unilemilar vesicles (LUVs) were made by tip-sonication using a lipid mixture of DOPC/sphingomyelin/cholesterol/DHPE-Rhodamine B (L1392, ThermoFisher)/PIP2(P-4516, Echelon) (30:40:10:10:10) in PBS buffer solution. Samples contained purified GFP-tagged Moesin protein (1 μ M) in the buffer (150mM NaCl and 10mM HEPES). PIP2-containing LUVs (10% PIP2) were added into the protein solution. PIP2 here functions as docking sites for Moesin. When Moesin binds to PIP2, the C-terminal tagged GFP (FRET donor) will be brought close to the vesicle surface where Rhodamine B (FRET acceptor)-labeled DHPE molecules reside closely. The protein to membrane FRET was measured with a spectrofluorimeter (FLS980 Spectrometer, Edinburgh Instruments) at 25 °C, with both excitation and emission slits at 2 nm. The samples were excited at 488 nm and the emission from 500-700 nm was collected for FRET analysis. To correct for direct acceptor excitation, we subtracted the background rhodamine B fluorescence of LUVs only samples from the experimental samples. In addition, purified GFP only was probed for membrane docking as well for negative control.

4. Phylogenetic trees shown in Fig. 4 are intriguing, though analysis of additional ITAM-containing proteins is needed to strengthen the authors assumption.

We have added CD3 zeta chain and DAP12 (Fig. 4; resemblance to gamma chain for FcRs by some analyses). Among the genes with ITAM, these two are crucial for immune tyrosine signaling and can be argued to account for a big percentage of the signaling events downstream of modern immune receptors in T cells and NK cells. B cells and phagocytes use Syk that is already included in the original manuscript. Note: we ask for the editor's attention again regarding the placement of this figure vs the potential use of the abbreviated version in this place.

5. Authors should clearly state the purpose of using MSU in the experiment in the main text.

We presume the reviewer is asking why some experiments MSU is used instead of beads. MSU is a stronger phagocytic target. We in previous work have demonstrated that for both targets, similar activation pathways were used. For timed experiments and for surface labeling, beads are more suited. However, to trigger a strong signaling cascade, MSU creates a more uniform and robust response. (insert 7)

Insert 7. Downstream signaling of Syk in particle phagocytosis

DC2.4 cells were treated with MSU, Silica crystal or Latex beads for indicated times. Total cell lysates were subjected to immunoblotting with the indicated antibodies.

6. Fig3a, b suggests that a functional ITAM can trigger phagocytosis, independent of Moesin. However, the phosphorylation of the ITAM is still required for this particular Moesin-independent phagocytic process. What drives the phosphorylation? It is known that Lck is responsible for the phosphorylation of ITAM in CD3 and ζ chains in the TCR. Is it possible that Lck phosphorylation might be occurring in this scenario? Is this phenomena only specific to the CD4 chimera or can other surface molecules be converted into ITAM chimeras and this sequence of events repeated?

Of Src family kinase, Lck is certainly well characterized by its association with CD4 polypeptide and its expression is limited to T cells. We therefore presume that other Src family members must be involved. In addition, constructs with ICAM and CD8 as the extracellular binding target would argue that CD4 association with Lck is not essential for our mode of phagocytosis (Fig. S3d). With that said, ITAM phosphorylation is primarily the act of Src family members (including Fyn, Hck, Fgr, Yes etc) that have less preferential cell type specificity. Here we show that incubation with Src family kinase inhibitors PP2 and AZD0530 reduced the phagocytosis (Fig. S3b). Of note, this result does not mean other PTKs are not involved as they most likely do as indicated by the residual activities under Src suppression.

7. The authors switch between DC2.4 (fig S3d) to RAW264.7 cells (fig S3e). Is there a reason for this?

In our efforts to make better Moesin KD, DC2.4 was proven to be better target. However, for most assays, particularly for high quality imaging, RAW264.7 cells are superior and more amenable to transfection by Xfect, a special reagent designed to transfect RAW264.7 cells with good efficiency, and are thus more commonly used (Schlam et al., 2015; Sezgin et al., 2012; Wong et al., 2016).

8. Fig S3h. The authors switch to murine BMDM. Is this change specific with regard to SR-A6/SR-A1 experiments? If so, why?

Yes, we received as a gift from Dr. Dawn Bowdish the SR KO mice. To match the cell types, we needed to produce similar macrophages for fair comparison.

Minor comments

1. Fig 1c. Seven clones were shown in the immunoblot. However, the phagocytosis assay shows only two. Which clones were used?

Out of several analyses, two clones showing the highest knockdown efficiency were used. We forgot to mention this in the last version. This has been added to the figure legend.

2. Lines 72-75, citations are required.

Revised

3. Fig 2d, S2b. PH-mCherry not mentioned in the Materials and Methods section. Authors should explain where this plasmid was obtained from, indicating the catalogue number. If the authors generated the plasmid, it should also be included in the Materials and Methods section. It should be noted that in general, the PH-domain can bind both PIP2 and PIP3. Therefore, it should be specified how specific the PH-domain is to the former and latter.

We obtained PH-mCherry from Addgene (Plasmid# 36075). The PH domain used in this construct is from PLCdelta, which preferentially binds to PIP2 over PIP3 as opposed to the PH domain of Akt (which preferentially binds to PIP3). We revise the name of the plasmid to PH-PLC δ -mCherry to provide better clarity (Chisari et al., 2009).

Reviewer #3 (Remarks to the Author):

Mu et al describe the finding that the actin binding protein, Moesin, functions to activate the Syk tyrosine kinase during uptake of solid particles by phagocytic cells. Specifically, interaction of particles with the external plasma membrane (either directly or through receptor-mediated binding) leads to accumulation of PIP2 in the inner portion of the plasma membrane, resulting in activation of Moesin, which through an ITAM-like sequence associates with Syk kinase to mediate downstream signaling leading to actin polymerization and uptake of the particle. Knockdown of Moesin reduces solid particle (latex bead) uptake; re-expression of the Moesin FERM domain (which has the ITAM like sequence) rescues particle uptake, while expression of an ITAM-point mutant of the Moesin FERM domain is inactive. PIP2, Moesin and Syk kinase all co-localize around the particle during uptake. Moesin-mediated signaling of may also be involved in scavenger receptor mediated particle uptake.

This is an interesting and well done paper. The role of ERM proteins linking to actin to various receptors is well described, but the idea that Moesin is signaling intermediate is novel and interesting. There are several questions that remain to be addressed.

We thank the reviewer for the positive comments.

1) Moesin knockdown in DC2.4 cells reduces latex bead uptake from ~80% to ~30%. Cyto D treatment goes to ~5%. What accounts for the remaining bead uptake? What is the efficiency of reduction of bead uptake with Syk knockdown in the same cells? Would this experiment be done better with CRISPR deletion of Moesin?

First, the 5% under CytoD treatment is probably background noise indicating perhaps random quenching of fluorescence by cellular mass. The remaining uptake above 5% is likely the incomplete KD by viral vector based iRNA delivery or the result of other ERM proteins mediating smaller scale phagocytosis. Regarding whether this can be done with a more thorough KD or KO, in our answer to reviewer 1, the complete reduction of Moesin will render cells detached from the plate. This makes the phagocytosis assay difficult (we need to observe the stationary attachment of beads to the cells as the baseline). To compare Moesin KD with Syk KD, we took the advice and carried out the experiment. Here, Moesin KD and Syk KD had similar reduction in phagocytosis (Fig. S11). As Syk is required for phagocytosis, we believe the result may also indicate the incomplete Syk blockage. If phagocytosis were entirely explained by our model, ideally, if Moesin/Syk were knocked out completely, the phagocytosis would have been reduced significantly. However, this is a risky assumption. First, the technical issues in our analyses do not give us the confidence to make a such statement. Second, although we believe that the lipid based signaling cascade described here may be an important mechanism, phagocytosis can be additionally explained by other mechanisms, such as adhesion molecule based (Dupuy and Caron, 2008; Dustin, 2016). In addition, Syk is myeloid molecule, yet some phagocytosis can take place in fibroblasts and epithelial cells, clearly indicating other mechanisms unrelated to our model. As these events can be numerous and hard to track down one by one, we have revised the manuscript to clear state the alternative possibilities, and suggest that our model, while explaining lipid sorting based phagocytosis, does not explain all the cellular uptake of particulate substances.

2) I was very surprised that transfection of the FERM domain of Moesin alone could rescue. How is that possible, since there would be no way to link polymerization of the actin cytoskeleton to the plasma membrane. Do the authors have some idea why this works? Some discussion of this is needed.

In light of our data, one potential way to think about it is that phagocytosis requires a concentrated accumulation of ITAM motifs at the targeted spot on the cell membrane. Most of the time, as in the case of FcR, ITAM is provided in the cytoplasmic domain (FcRIIA) or a common signaling chain (all other FcRs except for FcRIIB, which has an ITIM). Without receptor, lipid sorting results in the accumulation of PIP2 at the spot that sense the physical force. PIP2 at sufficient quantity recruits Moesin to the membrane. There, Moesin has two functions. 1. it provides the ITAM sequence to attract Syk, and 2. it stabilizes the link between inner leaflet to the cortical cytoskeleton, which may be important to anchor phagocytic machinery. In the absence of the second function, Syk recruitment alone is sufficient to induce downstream signaling, the membrane/cytoskeleton attachment can be mediated by other linker proteins (other ERMs, band 4.1 like protein, spectrin etc). The latter may be the primary mechanism for FcR mediated phagocytosis as in that case ITAM sequence is not linked to a cytoskeletal anchor. This possibility is discussed in this version.

3) Most importantly, there is no data or discussion of phosphorylation of Moesin. There is a phosphorylation event shown in the summary model, but it is not demonstrated in the data. Does latex bead binding to the membrane induce phosphorylation of Moesin? Is this mediated by Src-family kinases? How would they sense association of a latex bead to the plasma membrane? Some experiments examining the kinetics of Moesin phosphorylation are needed.

Thus far there is no phos-Moesin (ITAM Y191/Y205) specific antibody, which would be required for a direct analysis. With that said, two data points may suggest that this is the case. In Fig. 1e, we can clearly see that Y mutation to F renders the FERM domain ineffective. This is analogous to how immune tyrosine work in other ITAMs, and we believe it should not be an exception in our case. We added another experiment to show that in Moesin based phagocytosis, Src inhibitors blocked the uptake,

suggesting Src family kinases are required (Fig. S3b). Due to lack of reagents, we cannot at this moment carry the kinetics experiments proposed by the reviewer, we instead provide a discussion to cover this matter. Regarding how Moesin senses latex bead binding, the same issue has been raised by reviewer 2. Please see our answer to that question.

4) Why are Cos-1 and Cos-7 cells phagocytic but HEK293T and NIH3T3 not? Is it expression of Syk?

We do not know the exact answer. The reviewer is basically asking the question what the defining molecular signatures are for a phagocyte. This is a historical question yet without a clear answer. Beyond the requirement of Syk, we do not know what these components are, nor does the field. In the process of carrying out the project, we made attempts to extract some answer, to no avail. However, there is an interesting point that may be of value. Insert 8 shows that DC2.4 and RAW cells have clear Syk expression. Cos cells on the other hand expressed Zap70. There are reports that suggest that while Syk is the dominant player in phagocytosis, Zap70 can in some cases signal in place of Syk (Cheng et al., 1997; Greenberg et al., 1996).

Insert 8. ERM and Syk/Zap-70 expression level in different cell types.

Total lysate of DC2.4, RAW264.7, Cos-1, Cos-1, HEK-293T and NIH3T3 cells were subjected to immunoblotting with the indicated antibodies.

5) Most importantly, I believe that Grinstein's lab demonstrated many years ago that during FcR-mediated phagocytosis that PIP2 is EXCLUDED from the base of the phagocytic cup to allow actin polymerization of the arms of membrane around the IgG opsonized particle. The authors should discuss why PIP2 accumulation is required for inert particle uptake and seems to be excluded for FcR mediated phagocytosis.

Yes, this is a quintessential observation in phagocytosis. The internalization of phagocytic target is accompanied by dissolution of PIP2 and actin as well as conversion of ceramide signalling from membrane sphingolipids, much of it is associated with lipid signaling along with the phagocytosis development. In (Botelho et al., 2000; Scott et al., 2005) as well as in our own observation (insert 9, also see the link for our movie online goo.gl/e5qee1), the PIP2 and actin exclusion happens after the internalization (visually, when the positive curvature is bent inward with the deeper insertion of phagocytic cup). This is shared feature for both lipid sensing based and FcR based phagocytosis. We offered a short discussion in this version.

Insert 9. Dynamics change of PIP2 activation during solid particle phagocytosis

PH-PLC δ -GFP and LifeAct-tdTomato were co-expressed in RAW 264.7 cells. Cells were incubated with naked 3 μ m polystyrene beads at 37 °C for 40 minutes. Images were taken at a 5s interval for 40 minutes. The image above displays the spatial distribution of PIP2 and actin around the beads at different stages of phagocytosis before phagosome sealing. The bead of interest was indicated with '*' in yellow. (a) During initial contact of the bead to the plasma membrane, PIP2 and actin accumulate at the base of a phagocytic cup. (b) As phagocytosis progresses, PIP2 and actin start to accumulate around more in the membrane in contact with the bead. (c) At later stages of phagocytosis, PIP2 accumulation disappeared alongside actin in most areas of cell membrane enclosing the bead except for the top area. N=3. Scale bar, 5 μ m. The complete video can be accessed at goo.gl/e5qee1.

6) I believe that MSU uptake by macrophages is very pro-inflammatory while latex bead uptake is not. Is there a difference in the proximal signaling responses (PIP2 accumulation, Syk activation, Moesin phosphorylation) that may help explain this?

This is true. We have unpublished data that show the strong TNF alpha induction by MSU but not by latex beads. For reasons that we still do not understand, MSU appears to trigger stronger activation of pAKT, ERK and p38 (insert 7). On single cell force spectroscopy that we used routinely in the lab, MSU also exerts a stronger binding force (our own observation). We may speculate that MSU surface chemistry may result in a more efficient lipid sorting (we recently published a review related to the topic (Shu and Shi, 2018)). Additionally, the event of MSU internalization may trigger stronger inflammation in an uncharacterized manner.

Otherwise this is an interesting report that helps explain the mechanisms of inert particle engulfment by phagocytic cells.

References

- Ben-Aissa, K., Patino-Lopez, G., Belkina, N.V., Maniti, O., Rosales, T., Hao, J.-J., Kruhlak, M.J., Knutson, J.R., Picart, C., and Shaw, S. (2012). Activation of Moesin, a protein that links actin cytoskeleton to the plasma membrane, occurs by phosphatidylinositol 4,5-bisphosphate (PIP₂) binding sequentially to two sites and releasing an autoinhibitory linker. *J Biol Chem* 287, 16311-16323.
- Botelho, R.J., Teruel, M., Dierckman, R., Anderson, R., Wells, A., York, J.D., Meyer, T., and Grinstein, S. (2000). Localized biphasic changes in phosphatidylinositol-4,5-bisphosphate at sites of phagocytosis. *J Cell Biol* 151, 1353-1368.
- Bretscher, A., Edwards, K., and Fehon, R.G. (2002). ERM proteins and merlin: integrators at the cell cortex. *Nat Rev Mol Cell Biol* 3, 586-599.
- Cheng, A.M., Negishi, I., Anderson, S.J., Chan, A.C., Bolen, J., Loh, D.Y., and Pawson, T. (1997). The Syk and ZAP-70 SH2-containing tyrosine kinases are implicated in pre-T cell receptor signaling. *Proc Natl Acad Sci U S A* 94, 9797-9801.
- Chisari, M., Saini, D.K., Cho, J.H., Kalyanaraman, V., and Gautam, N. (2009). G protein subunit dissociation and translocation regulate cellular response to receptor stimulation. *PLoS One* 4, e7797.
- Dupuy, A.G., and Caron, E. (2008). Integrin-dependent phagocytosis: spreading from microadhesion to new concepts. *J Cell Sci* 121, 1773-1783.
- Dustin, M.L. (2016). Complement Receptors in Myeloid Cell Adhesion and Phagocytosis. *Microbiol Spectr* 4.
- Edwards, S.D., and Keep, N.H. (2001). The 2.7 Å crystal structure of the activated FERM domain of Moesin: an analysis of structural changes on activation. *Biochemistry* 40, 7061-7068.
- Golovkina, K., Blinov, A., Akhmametyeva, E.M., Omelyanchuk, L.V., and Chang, L.S. (2005). Evolution and origin of merlin, the product of the Neurofibromatosis type 2 (NF2) tumor-suppressor gene. *BMC Evol Biol* 5, 69.
- Greenberg, S., Chang, P., Wang, D.C., Xavier, R., and Seed, B. (1996). Clustered syk tyrosine kinase domains trigger phagocytosis. *Proc Natl Acad Sci U S A* 93, 1103-1107.
- Hamada, K., Shimizu, T., Matsui, T., Tsukita, S., and Hakoshima, T. (2000). Structural basis of the membrane-targeting and unmasking mechanisms of the radixin FERM domain. *EMBO J* 19, 4449-4462.
- Hao, J.-J., Liu, Y., Kruhlak, M., Debell, K.E., Rellahan, B.L., and Shaw, S. (2009). Phospholipase C-mediated hydrolysis of PIP₂ releases ERM proteins from lymphocyte membrane. *J Cell Biol* 184, 451-462.
- Hirao, M., Sato, N., Kondo, T., Yonemura, S., Monden, M., Sasaki, T., Takai, Y., Tsukita, S., and Tsukita, S. (1996). Regulation mechanism of ERM (ezrin/radixin/Moesin) protein/plasma membrane association: possible involvement of phosphatidylinositol turnover and Rho-dependent signaling pathway. *J Cell Biol* 135, 37-51.
- Johnson, C.M., and Rodgers, W. (2008). Spatial Segregation of Phosphatidylinositol 4,5-Bisphosphate (PIP₂) Signaling in Immune Cell Functions. *Immunology, endocrine & metabolic agents in medicinal chemistry* 8, 349-357.
- Kim, H.M., Choo, H.J., Jung, S.Y., Ko, Y.G., Park, W.H., Jeon, S.J., Kim, C.H., Joo, T., and Cho, B.R. (2007). A two-photon fluorescent probe for lipid raft imaging: C-laurdan. *Chembiochem* 8, 553-559.
- Lai, C.L., Landgraf, K.E., Voth, G.A., and Falke, J.J. (2010). Membrane docking geometry and target lipid stoichiometry of membrane-bound PKC α C2 domain: a combined molecular dynamics and experimental study. *J Mol Biol* 402, 301-310.
- Li, Q., Nance, M.R., Kulikaukas, R., Nyberg, K., Fehon, R., Karplus, P.A., Bretscher, A., and Tesmer, J.J. (2007). Self-masking in an intact ERM-merlin protein: an active role for the central alpha-helical domain. *J Mol Biol* 365, 1446-1459.

Lubart, Q., Vitet, H., Dalonneau, F., Le Roy, A., Kowalski, M., Lourdin, M., Ebel, C., Weidenhaupt, M., and Picart, C. (2018). Role of Phosphorylation in Moesin Interactions with PIP2-Containing Biomimetic Membranes. *Biophys J* *114*, 98-112.

Machnicka, B., Czogalla, A., Hryniewicz-Jankowska, A., Bogusławska, D.M., Grochowalska, R., Heger, E., and Sikorski, A.F. (2014). Spectrins: A structural platform for stabilization and activation of membrane channels, receptors and transporters. *Biochimica et Biophysica Acta (BBA) - Biomembranes* *1838*, 620-634.

Pearson, M.A., Reczek, D., Bretscher, A., and Karplus, P.A. (2000). Structure of the ERM protein Moesin reveals the FERM domain fold masked by an extended actin binding tail domain. *Cell* *101*, 259-270.

Ponuwai, G.A. (2016). A glimpse of the ERM proteins. *Journal of Biomedical Science* *23*, 35.

Schlam, D., Bagshaw, R.D., Freeman, S.A., Collins, R.F., Pawson, T., Fairn, G.D., and Grinstein, S. (2015). (SuppInfo) Phosphoinositide 3-kinase enables phagocytosis of large particles by terminating actin assembly through Rac/Cdc42 GTPase-activating proteins. *Nature communications* *6*, 8623-8623.

Scott, C.C., Dobson, W., Botelho, R.J., Coady-Osberg, N., Chavrier, P., Knecht, D.A., Heath, C., Stahl, P., and Grinstein, S. (2005). Phosphatidylinositol-4,5-bisphosphate hydrolysis directs actin remodeling during phagocytosis. *J Cell Biol* *169*, 139-149.

Sezgin, E., Kaiser, H.J., Baumgart, T., Schwille, P., Simons, K., and Levental, I. (2012). Elucidating membrane structure and protein behavior using giant plasma membrane vesicles. *Nat Protoc* *7*, 1042-1051.

Shi, Y. (2012). To forge a solid immune recognition. *Protein Cell* *3*, 564-570.

Shu, F., and Shi, Y. (2018). Systematic Overview of Solid Particles and Their Host Responses. *Front Immunol* *9*, 1157.

Urzainqui, A., Serrador, J.M., Viedma, F., Yanez-Mo, M., Rodriguez, A., Corbi, A.L., Alonso-Lebrero, J.L., Luque, A., Deckert, M., Vazquez, J., *et al.* (2002). ITAM-based interaction of ERM proteins with Syk mediates signaling by the leukocyte adhesion receptor PSGL-1. *Immunity* *17*, 401-412.

Vachon, E., Martin, R., Plumb, J., Kwok, V., Vandivier, R.W., Glogauer, M., Kapus, A., Wang, X., Chow, C.W., Grinstein, S., *et al.* (2006). CD44 is a phagocytic receptor. *Blood* *107*, 4149-4158.

Wong, H.S., Jaumouillé, V., Freeman, S.A., Doodnauth, S.A., Schlam, D., Canton, J., Mukovozov, I.M., Saric, A., Grinstein, S., and Robinson, L.A. (2016). Chemokine Signaling Enhances CD36 Responsiveness toward Oxidized Low-Density Lipoproteins and Accelerates Foam Cell Formation. *Cell Reports* *14*, 2859-2871.

Yonemura, S., Matsui, T., Tsukita, S., and Tsukita, S. (2002). Rho-dependent and -independent activation mechanisms of ezrin/radixin/Moesin proteins: an essential role for polyphosphoinositides in vivo. *J Cell Sci* *115*, 2569-2580.

REVIEWERS' COMMENTS:

Reviewer #1 (Remarks to the Author):

I believe that the authors have provided a very thoughtful and complete response to my original concerns. I recommend publication of this very interesting manuscript

Reviewer #2 (Remarks to the Author):

The revised manuscript is substantially improved.
This manuscript is ready for publication.

Reviewer #3 (Remarks to the Author):

This is a revised version of fine manuscript that defines the mechanisms of inert particle phagocytosis. The paper was very well received by all three reviewers and their questions/suggestions were all well addressed in the revision. The revision has a number of new experiments and text changes that improve the paper. This is a strong revision of a strong paper. Congratulations to the authors.